# Multiple technical observations of the atmospheric boundary layer structure of a red warning haze episode in Beijing

Yu Shi[1,2], Fei Hu[1,2], Guangqiang Fan[3], and Zhe Zhang[1,2]

[1]State Key Laboratory of Atmospheric Boundary Layer Physics and Atmospheric Chemistry, Institute of Atmospheric Physics, Chinese Academy of Sciences, Beijing 100029
[2]University of Chinese Academy of Sciences, Beijing 100049
[3]Key Laboratory of Environmental Optics and Technology, Anhui Institute of Optics and Fine Mechanics, Chinese Academy of Sciences, Hefei 230031

**Correspondence:** F.Hu (hufei@mail.iap.ac.cn)

**Abstract.** The study and control of air pollution involves measuring the structure of the atmospheric boundary layer (ABL) to understand the mechanisms of the interactions occurring between the atmospheric boundary layer and air pollution. However, when extreme pollution occurs, the detection of atmospheric boundary layer structures is very limited. Beijing, the capital of China, experienced severe levels of haze pollution in December 2016, and the city issued its first red air pollution warning of the year (the highest $PM_{2.5}$ concentrations were later found to exceed $450\mu g\ m^{-3}$). In this paper, the vertical profiles of wind, temperature, humidity and the extinction coefficient (reflecting aerosol concentrations) as well as ABL heights and turbulence quantities under heavy haze pollution conditions are analyzed, with data collected from Lidar, wind profile radar (WPR), radiosonde, a 325-meter meteorological tower (equipped with a 7-layer ultrasonic anemometer and 15-layer low frequency wind, temperature and humidity sensors) and ground observations. The ABL heights obtained by three different methods based on Lidar extinction coefficient data ($H_c$) are compared with the heights calculated from radiosonde temperature data ($H_\theta$) and from WPR wind speed data ($H_u$). The results show that increases in water vapor have greatly promoted the hygroscopic growth of aerosols and that corresponding extinction coefficient levels have also increased significantly. The ABL heights of $H_\theta$ and $H_u$ measured on days with heavy levels of haze pollution were generally lower than those measured on days without pollution, but the $H_c$ levels increased. Turbulence and pollutant concentrations are closely related during periods of haze pollution, and time changes in friction velocity ($u_*$) and turbulent kinetic energy (TKE) are clearly inversely correlated with $PM_{2.5}$ levels. The results in this paper can serve as a reference for the parameterization of boundary layer heights and turbulent diffusion processes for the numerical model of severe air pollution.

## 1 Introduction

Air pollution has an important impact on human health, weather, climatic patterns and the ecological environment (Seinfeld and Pandis, 1997; Brook et al., 2004; Ding et al., 2013; Wang et al., 2014a; Zhang et al., 2015a). The pollutants emitted as a result of human activities are mainly confined to the atmospheric boundary layer (ABL), which is the lowest part of the troposphere and is approximately 1∼2 km from the ground. In particular, fog and haze, which have a strong influence on visibility and air

quality levels, mainly occur in the ABL (Cao et al., 2004; Chan and Yao, 2008; Fu et al., 2008; Liu et al., 2012). Because the formation, evolution, and diffusion of air pollutants are closely related to ABL structures and turbulence characteristics (Zhang et al., 2011; Wei et al., 2018), research on the ABL is important for understanding air pollution mechanisms and for developing pollution control strategies. On the other hand, the relationships between the ABL and atmospheric pollution are very complex
and involve multiscale nonlinear physical and chemical processes; thus, both theoretical research and numerical simulations have encountered difficulties (Sun et al., 2013; Huang et al., 2014; Wang et al., 2014b; Miao et al., 2018). Therefore, it is very necessary to obtain first-hand information from observation experiments. Because of the Earth's rotation, the ABL presents strong diurnal variation, leading to the formation of many different layers in the boundary layer. The mixing layer accounts for a large proportion of the ABL in the deep convective boundary layer, and at present, the height of the mixing layer is equivalent
to the height of the ABL. Pollutants emitted into the ABL can reach a certain height through turbulent vertical mixing processes (Emeis and Schäfer, 2006), making it possible to determine the ABL height from the concentration of pollutants. The top of the mixing layer exhibits capping inversion. Due to a change in the surface net radiation occurring at night, a stable boundary layer begins to form at night because of the cooling effect of the ground surface, and the surface inversion layer is nearest to the ground. The nocturnal stable boundary layer is often accompanied by a residual layer that maintains the characteristics of the
daytime mixing layer (Stull, 1988). The ABL height is closely related to air pollution, but it is not the only factor that shapes air quality. Pollution conditions are also affected by wind speeds, emissions, chemical processing, etc. (Schäfer et al., 2006; Gei$\beta$ et al., 2017). Some works have compared ABL heights based on lidar and radiosonde data, and the correlations between them are stronger under unstable conditions (Emeis and Schäfer, 2006; Martucci et al., 2006). However, for ABL heights determined from wind profiler radar (WPR), relatively fewer studies have compared and analyzed the results of lidar and radiosonde tests
applied during haze pollution episodes.

Regarding air pollution, many observational experiments have been conducted internationally, especially with reference to air pollution in the ABL over urban areas (i.e., the urban boundary layer). Examples of such projects include European Cooperation in the Field of Scientific and Technical Research, abbreviated as COST715 (Fisher et al., 2001); URBAN 2000, a major urban tracer and meteorological field campaign conducted in Salt Lake City, Utah, in October 2000 (Allwine et al.,
2002); Joint Urban 2003, a field experiment conducted in October 2003 in Oklahoma City (Wang et al., 2007); MIRAGE 2006, Megacity Impacts on Regional and Global Environments (Lance et al., 2012); and SURF, the Study of Urban impacts on Rainfall and Fog/haze (Liang et al., 2018).

A meteorological tower serves as one of the best platforms from which to detect the ABL structure under conditions of atmospheric pollution (Quan and Hu, 2009; Sun et al., 2015; Ren et al., 2018). Although the height of such a tower is limited,
the boundary layer is basically stable when heavy pollution occurs, and the ABL height is low, so it is easy to measure from a tower. Conventional meteorological and turbulence instruments installed at different heights above a meteorological tower can obtain information on stable boundary layer structures and turbulence diffusion parameters (Katul et al., 1995). Traditional detection methods include tethered balloon, radiosonde, WPR tools, which can detect higher heights (Grimsdell and Angevine, 1998; Andreas et al., 2000; Kalapureddy et al., 2007; Li et al., 2015; Han et al., 2018). In recent decades, aerosol laser
radar (Lidar) has been used increasingly extensively. It can be used to retrieve the vertical distribution of particles from Lidar

backscattering data (Wang et al., 2012; Summa et al., 2013; Jiannong et al., 2013; Bravo-Aranda et al., 2017). It is impossible to obtain information on the boundary layer structure and on the interrelationships between pollutants found in atmospheric pollution (especially in heavy haze) unilaterally by means of the above-mentioned technical techniques, and it is necessary to carry out comprehensive observations simultaneously.

From 14 to 22 December 2016, Beijing, the capital of China, experienced a period of severe haze pollution. The government issued its highest air pollution warning (red alert) during this period. Beijing is a densely populated city covering an area of approximately 396 square kilometers (see Fig.1b). Despite strong pollution control measures taken by the government, the average $PM_{2.5}$ concentration per hour rose from $20\mu g$ m$^{-3}$ to more than $450\mu g$ m$^{-3}$ (see Table.1) in just five days. What are the mechanisms of episodes of such severe air pollution? Addressing this question requires conducting a comprehensive and

in-depth analysis of weather conditions, pollutant emissions, regional transport processes and physicochemical transformation mechanisms and of interactions between haze and boundary layer structures (Huang et al., 2014; Sun et al., 2014; Ding et al., 2016). Some previous studies have been conducted on haze events in the Beijing area (Li et al., 2017; Sheng et al., 2018; Wang et al., 2018), especially physical and chemical mechanistic analyses based on observation data for large towers with multiple features (Sun et al., 2006; Guo et al., 2016).

The purpose of this paper is to investigate the ABL's structure and turbulence characteristics measured during the red haze warning period of 2016 by means of tower, Lidar, WPR and radiosonde approaches. The paper includes a brief introduction of weather patterns, of heavy haze pollution trends and of our observation sites and techniques; an analysis of boundary layer winds, temperatures, humidity profiles, and extinction coefficients (reflecting the concentration of haze particles); and ABL heights determined by different detection techniques. The vertical distributions of turbulent quantities are also outlined. Finally,

avenues for further research are given.

## 2   Observation sites, instruments and data

The ABL observation data used for this paper mainly cover three locations in Beijing. The first area is located at the Institute of Atmospheric Physics (IAP) of the Chinese Academy of Sciences, where there are a 325 meter high meteorological tower and a Lidar. The second is positioned approximately 600 meters away from the east side of this tower where a wind profile radar

(WPR) system is based. The third area is the observatory of the Beijing Meteorological Bureau, which is approximately 20 kilometers away from the tower. Conventional ground meteorological observations and radiosonde data from the WMO station are used (ZBAA in Fig.1b). The above observation sites are shown in Figure 1b. The topography around Beijing is also given in Figure 1a. We use local station time in this work, and the observational instruments and data employed are as follows:

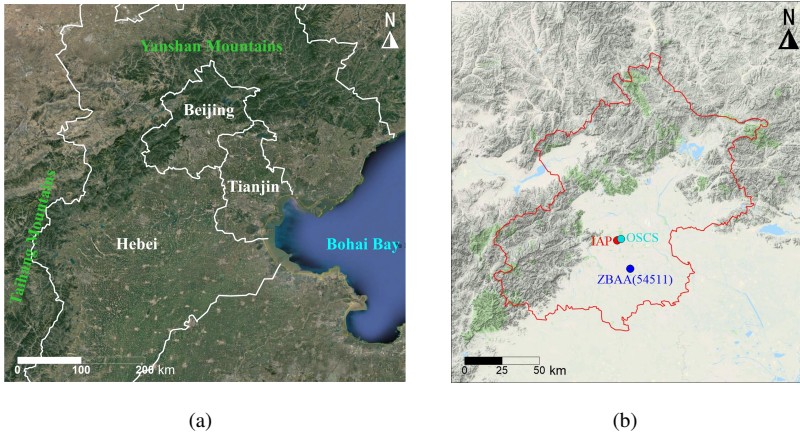

(a)            (b)

**Figure 1.** Local topography of Beijing and of its surrounding area (a). The locations of observation sites in Beijing (b): red circle: IAP (Lidar), blue circle: ZBAA radiosonde observation station, cyan circle: pollution observation station (OSCS) positioned approximately 2 km northeast of the Lidar. Beijing is a densely populated city covering an area of approximately 396 square kilometers.

    1) The IAP's meteorological tower is positioned 49 meters above sea level, is 325 meters tall, and is located at (39°58'N, 116°22'E) between the Beijing North Third Ring Road and North Fourth Ring Road. A total of 15 observation platforms (at 8, 15, 32, 47, 65, 80, 103, 120, 140, 160, 180, 200, 240, 280 and 320 m) are set up on the tower, and wind speed (MetOne, USA), wind direction (MetOne, USA), temperature (HC2-S3,Switzerland) and humidity (HC2-S3, Switzerland) observation instruments are mounted onto each platform. In addition, 7 sets of three-dimensional ultrasonic anemometers (Wind Master, Gill, USA) and water vapor / carbon dioxide analyzers (LI-7500,USA) are installed on the tower (at 8, 15, 47, 80, 140, 200 and 280 m). All turbulence data sampling frequencies are set to 10 Hz. All of the tower data are averaged for 20 minutes. A detailed description of the meteorological tower can be found in (Al-Jiboori and Fei, 2005) and (Chen et al., 2018) and on the website (http://view.iap.ac.cn:8080/imageview/).

    2) The extinction coefficients were measured by a Lidar system (AGHJ-I-Lidar, China) installed underneath the 325 m tower. The Lidar can provide backscattering signals at wavelengths of 532 nm and 355 nm at a vertical resolution of 7.5 m and a temporal resolution of approximately 5~10 min. Due to technical failures, lidar data are missing for 11:00 on December 19 to 09:00 on December 20, 2016;

    3) Wind speeds and wind directions were also monitored by means of WPR (Airda3000,China) during red-alert pollution periods. In this paper, the temporal resolution of WPR is set to 5 min, and the vertical resolution is set to 50 m below 1000 m and 90 m above 1000 m.

    4) High-resolution vertical profile radiosonde data collected twice daily (08:00 and 20:00 Beijing time) were retrieved from the University of Wyoming's website(http://weather.uwyo.edu/) for Beijing's meteorological observatory station, which is named ZBAA in international code (Fig.1b). Surface visibility and other normal meteorological variables were routinely measured with a temporal resolution of half an hour in ZBAA.

5) Surface measurements of six kinds of air pollutants ($PM_{2.5}$, $PM_{10}$, $NO_2$, $SO_2$, CO and $O_3$) with a temporal resolution of one hour can be found on the official website of the Beijing Environmental Protection Agency (http://beijingair.sinaapp. com/). The data used in this paper were collected from the environmental monitoring station (Olympic Sports Center Station) positioned closest to the tower (approximately 2 km northeast).

## 3    Results and discussion

### 3.1    Surface observations of haze and meteorological conditions

From 14 to 22 December 2016, complete haze pollution was observed in the Beijing area (see Table 1). The generation, accumulation and elimination of $PM_{2.5}$ were recorded. We can see that from 20 to 21 December, the hourly average $PM_{2.5}$ concentration was almost maintained at approximately 400 $\mu$g m$^{-3}$ over 48 hours, which greatly exceeded the air pollution limits (i.e., 250 $\mu$g m$^{-3}$) set by China's State Environmental Protection Administration. Figure 2 shows the concentration time series for $PM_{2.5}$, wind speed and direction, temperature, relative humidity (RH), surface pressure and visibility for this period of heavy haze pollution .

**Table 1.** Daily average data for six major air pollutants in Beijing measured during a period of heavy pollution from 14 to 23 December 2016: $PM_{2.5}$,$PM_{10}$,$NO_2$,$SO_2$ and $O_3$ (units are $\mu$g m$^{-3}$); CO(mg m$^{-3}$). Data sources: http://beijingair.sinaapp.com/; according to the "Technical Specification for Air Quality Index(HJ 633-2012)" issued by China's National Environmental Protection Agency, based on $PM_{2.5}$ concentrations air pollution levels can be divided into five levels, i.e., good($0\sim75$ $\mu$g m$^{-3}$), slightly polluted ($75\sim115$ $\mu$g m$^{-3}$), moderately polluted ($115\sim150$ $\mu$g m$^{-3}$), heavily polluted ($150\sim250$ $\mu$g m$^{-3}$), and seriously polluted ($>250$ $\mu$g m$^{-3}$).

| Date | Air Quality | AQI index | $PM_{2.5}$ | $PM_{10}$ | $NO_2$ | $SO_2$ | CO | $O_3$ |
|------|-------------|-----------|------------|-----------|--------|--------|------|-------|
| 2016-12-14 | Good | 60 | 24 | 38 | 42 | 9 | 0.74 | 32 |
| 2016-12-15 | Good | 83 | 25 | 51 | 40 | 9 | 0.85 | 31 |
| 2016-12-16 | Slightly Polluted | 274 | 101 | 134 | 87 | 20 | 2.07 | 8 |
| 2016-12-17 | Heavily Polluted | 351 | 184 | 211 | 102 | 30 | 3.14 | 5 |
| 2016-12-18 | Seriously Polluted | 337 | 219 | 245 | 100 | 24 | 3.42 | 7 |
| 2016-12-19 | Seriously Polluted | 306 | 214 | 247 | 107 | 22 | 3.88 | 7 |
| 2016-12-20 | Seriously Polluted | 342 | 365 | 422 | 133 | 8 | 7.67 | 4 |
| 2016-12-21 | Seriously Polluted | 363 | 393 | 429 | 152 | 10 | 7.97 | 4 |
| 2016-12-22 | Moderately Polluted | 325 | 93 | 170 | 45 | 6 | 1.95 | 39 |
| 2016-12-23 | Good | 55 | 31 | 42 | 43 | 7 | 0.74 | 26 |

Generally, visibility serves as a representative index of air quality and atmospheric diffusion capacity (Zhang et al., 2015b). Figure 2 shows that when the concentration of $PM_{2.5}$ increased to high levels, visibility quickly deteriorated. Visibility on clean days was largely measured as greater than 10 km, and when the $PM_{2.5}$ concentrations reached approximately $200\sim300$

$\mu$g m$^{-3}$, the visibility decreased to 2~5 km. Even when PM$_{2.5}$ reached approximately 400 $\mu$g m$^{-3}$, visibility dropped sharply to 1 km or to hundreds of meters. The surface pressure results suggest that air pressure levels decreased from approximately 1035 hPa to 1023 hPa, and in general, Beijing was controlled by a weak high-pressure system during the pollution episode. The RH taken from ground observations shows significant diurnal variations and an obvious anti-correlation between RH and temperature. From 20 to 21 December, the diurnal variation in temperature and relative humidity in heavy pollution was greatly suppressed, and a further analysis of MODIS images (see Fig. 3) during this period shows that the pollution process was indeed accompanied by fog, while pollution formed in the south-central area of Hebei Province on 15 December 2016 and then spread across the whole Beijing-Tianjin-Hebei area on 18 December. Stratiform clouds appeared in areas surrounding Beijing on 21 December, but due to the high concentrations of pollutants (PM$_{2.5}$ values approaching 400 $\mu$g m$^{-3}$), mixed fog and haze appeared in Beijing. During the day, pollutants can scatter more solar radiation while the ground receives less solar radiation, leading to the suppression of diurnal variations in temperature and relative humidity on the ground (Gao et al., 2015). An increase in RH occurs due to a decrease in temperature but is also the result of a surge in water vapor. For example, in the early morning, temperature differences observed between 17 and 20 December were minor, and the RH on 17 December was approximately 80%, while the RH in the early morning of 20 December reached nearly 100%, indicating an increase in water vapor levels in the Beijing area at this time. The surface wind speed during the pollution episode fell to almost less than 2 m s$^{-1}$ and can be basically regarded as a stagnant weather system dominating ABL processes and resulting in poor air quality. From Fig. 2, we can see that there were cold fronts (strong NW winds) on both 15 December and 22 December, which advected pollutants away, resulting in good air quality. Between the fronts, PM$_{2.5}$ levels slowly increased, as the air was stagnant (weak and variable winds in between), and the pollutants that were emitted locally slowly built up. The wind direction also seemed to be cyclical on each day in response to local mountain valley circulation around Beijing (Hu et al., 2005). According to other studies, stronger northerly winds occurring in the winter are the main mechanisms through which pollution is removed, leading to good air quality (Sheng et al., 2018).

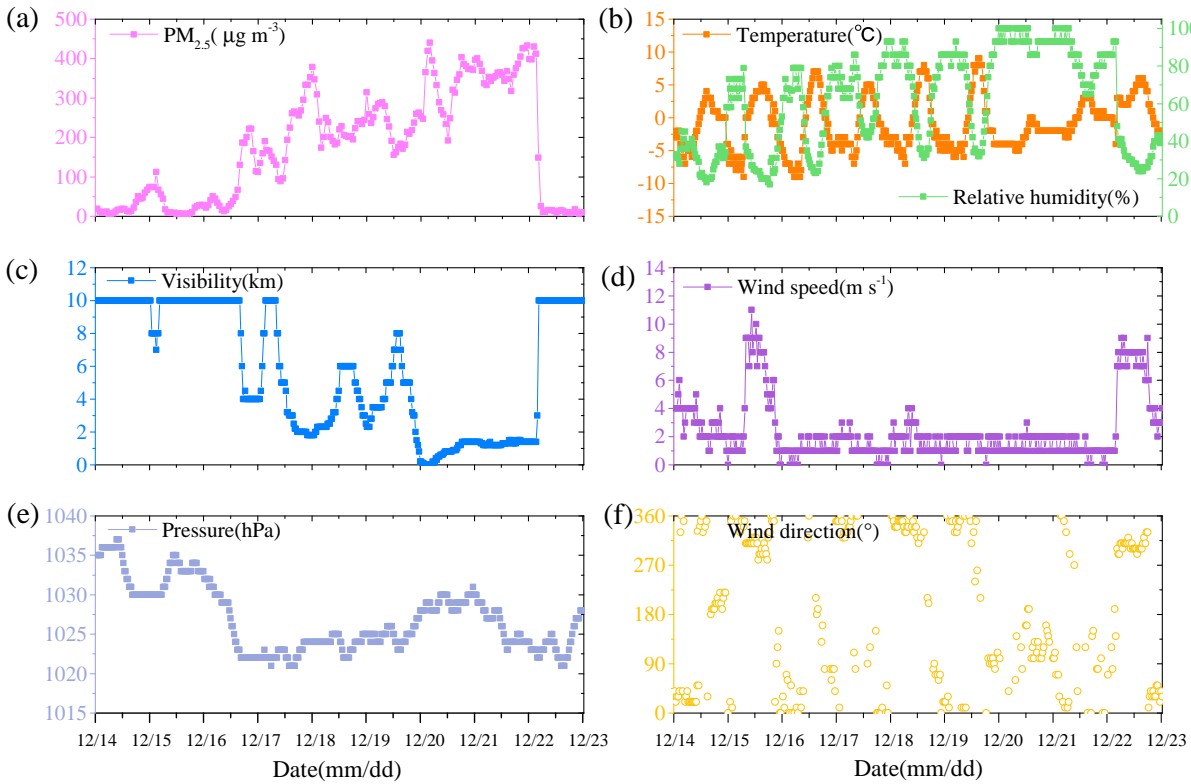

**Figure 2.** Time series of ground level PM$_{2.5}$ (a), relative humidity and temperature (b), visibility (c), wind speed (d), surface pressure (e) and wind direction (f) during 14 to 22 December 2016; the units for these meteorological parameters are as follows: $\mu$g m$^{-3}$, %,$^{\circ}$C, km, m s$^{-1}$, hPa and $^{\circ}$ respectively.

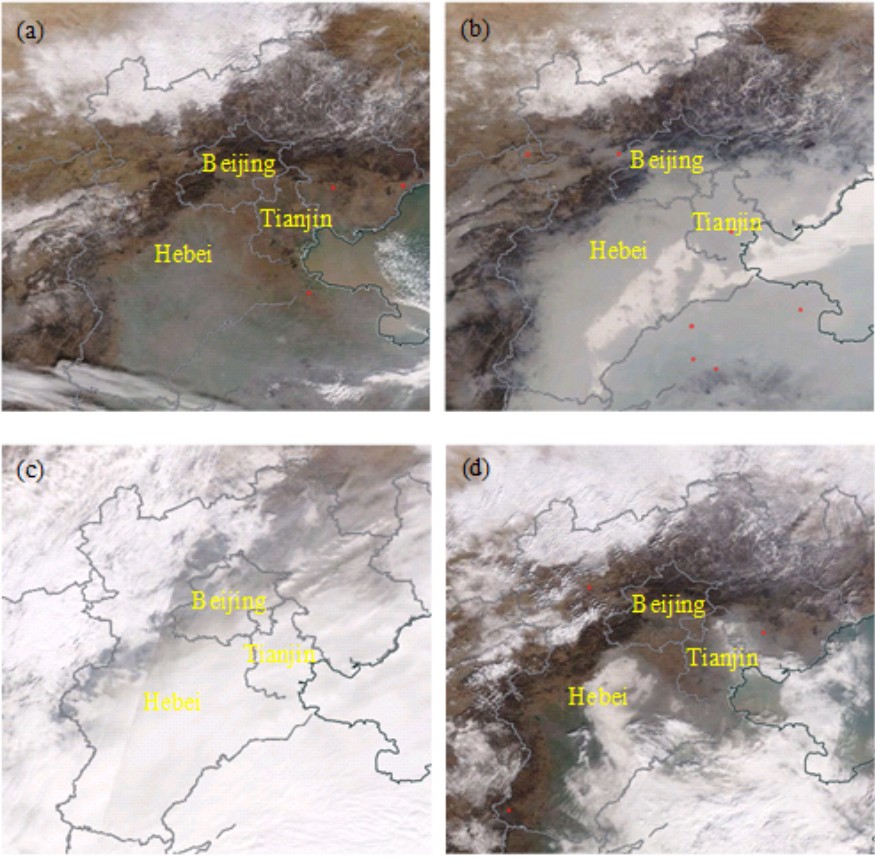

**Figure 3.** MODIS images of the Beijing-Tianjin-Hebei region on 15 December (a), 18 December (b), 21 December (c), and 22 December (d).

## 3.2 Boundary layer heights observed by Lidar

The most basic definition of the ABL height is the height at which the influence of the Earth's surface on the lower troposphere disappears. This influence applies not only to conventional meteorological elements but also to turbulence quantities and even more for substances in the atmosphere such as aerosols, water vapor and nonreactive tracer gases (Seibert et al., 2000). Levels
5  of various pollutants and water vapor in the ABL are much higher than those found in the free atmosphere, and therefore, there is often an obvious aerosol concentration gradient between the boundary layer and the free atmosphere. The extinction coefficient reflects the degree of aerosol particle scattering from lasers in the atmosphere (Boers and Eloranta, 1986). Thus, the ABL height can also be estimated from the extinction coefficient gradient. We used three popular methods—the gradient method (Lidar_gra) (Flamant et al., 1997), the standard deviation method (Lidar_std) (Hooper and Eloranta, 1986) and the
10  wavelet method (Lidar_wav) (Cohn and Angevine, 2000; Davis et al., 2000; Brooks, 2003)—to extract boundary layer heights from extinction coefficients. The ABL height determined by Lidar is represented by $H_c$. In this study, the Lidar_gra method

applies the height of the atmosphere at which the gradient of the Lidar extinction coefficient reaches its most negative value. The standard deviation of the extinction coefficient reflects the degree of Lidar echo signal dispersion at different heights. The top of the planetary boundary layer constitutes the intersection between air in the boundary layer and the free atmosphere, which leads to a strong signal change at the top of the boundary layer. We define the height of the maximum standard deviation of signals as the ABL height. The Lidar_wav method can also be used to detect abrupt changes in signals, so we use the Haar wavelet and take the height at which the wavelet coefficient is at its highest value as the height of the ABL. These methods are used to find the abrupt change in the extinction coefficient occurring at the top of boundary layer, though they present their own limitations.

Generally, the atmospheric boundary layer can be divided into a daytime convective mixing layer and a nighttime stable boundary layer. In the morning, the well-mixed convective boundary layer (CBL) is growing and often reaches its maximum height in the early afternoon. In the afternoon, the CBL gradually transforms into a neutral boundary layer. Figure 4 illustrates the evolution of ABL heights measured with Lidar, WPR and radiosonde tools.

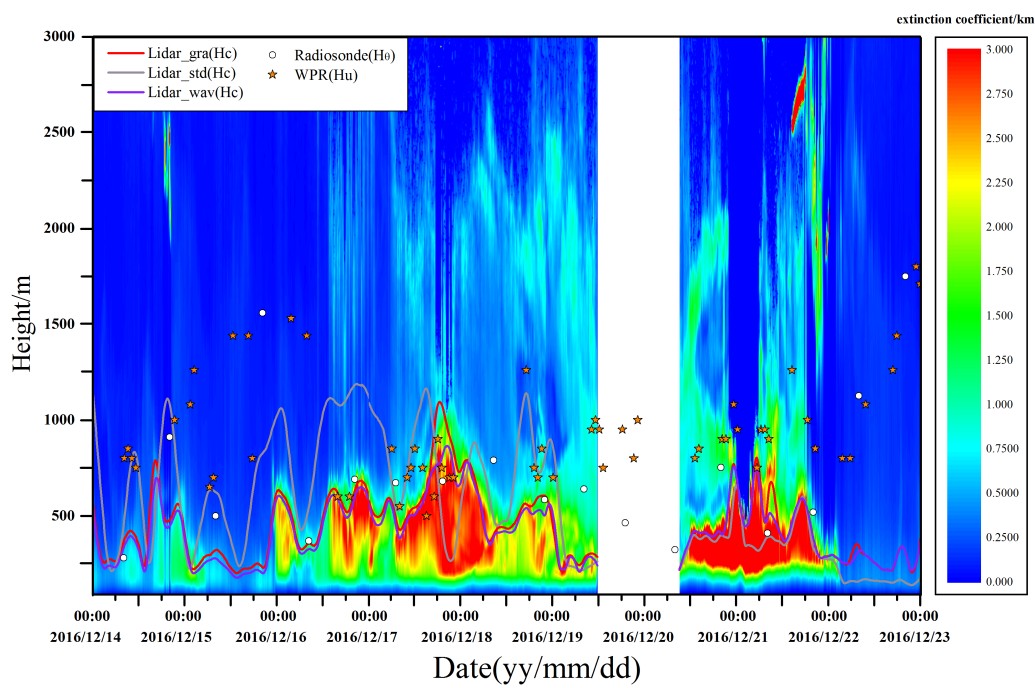

**Figure 4.** Temporal and spatial variations in the extinction coefficient (shaded, unit: km$^{-1}$) from 14 to 23 December 2016 and ABL heights (m) determined with different instruments. The red line (Lidar_gra), grey line (Lidar_std), and purple line (Lidar_wav) represent ABL heights determined by the Lidar using the gradient method, the standard deviation method, and the wavelet method, respectively. White points: ABL height determined by radiosonde; Five-pointed star: ABL height determined by WPR. It should be noted that the blank part of the extinction coefficient can be attributed to a technical failure, and lidar data for 11:00 on December 19 to 09:00 on December 20 are missing.

The determination of the ABL height by means of the Lidar method is based on the vertical profiles of the extinction coefficient or the aerosol concentration. When concentrations of PM$_{2.5}$ are high, the weakening effects of aerosol particles on lasers are stronger. ABL heights determined by Lidar_gra and Lidar_wav were almost the same, with a correlation coefficient of nearly 95%. From 16 to 18 December and from 20 to 21 December, the ABL heights were approximately 500~750 m. Further-5 more, the ABL height determined by the Lidar_std method was slightly higher than that derived from both methods. During a period of heavy pollution (20 to 21 December), the extinction coefficient quickly exceeded 3 km$^{-1}$ at 250 m aboveground. Perhaps due to the accumulation of pollutants, $H_c$ did not seem to decline on these days. When the atmosphere was relatively free of pollutants, such as on 15 or 22 December, the aerosol concentration was low and the extinction coefficient derived from the Lidar system displayed no obvious signs of decline from the ground to the upper height. The ABL heights obtained by 10 these methods based on the Lidar system are clearly lower than those obtained by the other instruments. If we were to continue

to use the Lidar system to determine the height of the boundary layer, it would have been distorted. Therefore, the continuous observation of the ABL height can be achieved by means of other instruments or improved methods based on the Lidar system.

### 3.3  Boundary layer structure observed by radiosonde technologies

Radiosonde instruments are the most widely used tools for conventional meteorological observation. The white points shown in figure 4 denote the ABL height determined from radiosonde data. The potential temperature ($\theta$) determined by radiosonde technology is calculated from the following formula: $\theta = T + \gamma_d z$. $\gamma_d = 0.00975$ K m$^{-1}$, and T is the measured temperature. As noted in many previous studies, the most widely used approach for the determination of the ABL height and structure for the daytime and nighttime involves identifying local maxima in potential temperature vertical gradient profiles as measured by radiosonde devices (Seibert et al., 2000; Summa et al., 2013; Sorbjan, 1989), but this method is only appropriate to apply to the convective ABL (Hennemuth and Lammert, 2006). Since the pollution period we examine involves stagnant winter weather conditions and as our radiosonde data only apply to dawn and dusk periods (08:00 h and 20:00 h), a stable boundary layer often appears (see Fig. 5, for example), and the height of the stable boundary layer (SBL) is more difficult to determine (Keller et al., 2010; Jong et al., 2015; Schäfer et al., 2006). In this study, the level showing an obvious change in the potential temperature gradient and the profile of relative humidity were used to define the ABL height, as expressed by $H_\theta$ (see the blue dotted lines in Fig. 5). Under stagnant and heavily polluted weather conditions, turbulence is more heavily suppressed than under normal weather conditions, and the top of the residual layer can also characterize the thickness of the stable boundary layer to some extent. We can also use the minimum value of the relative humidity (green curves shown in Fig. 5) gradient to determine the height of the SBL. The atmospheric stratification of potential temperatures and RH can affect the distribution of aerosol concentrations, which in turn affects the extinction coefficient. In Fig. 5, vertical profiles of the extinction coefficient observed by Lidar during the same period are also given.

    As shown in Fig. 5, the pollution episode was often accompanied by an inversion layer, as the vertical gradient of PT is positive, implying that the atmosphere was basically stable. The fact that the PBL was stable is unsurprising given the timing of radiosonde profiles for 08:00 and 20:00 local time, which respectively occur approximately 30 min after sunrise and 3 hours after sunset during the experimental period. Thus, a nocturnal inversion should barely be eroded by 08:00 (if at all, depending on the energy balance as insolation levels are low), and a nocturnal inversion should form by 20:00. Due to this timing, these profiles are not representative of daytime conditions when pollutants are actively mixed. Midday profiles (noon local time or the early afternoon) would instead present instability and mixing. Air pollutants are generally blocked below the inversion layer and are not easily diffused to high levels. Figure 5a shows that $H_\theta$ at 20:00 on 16 December was approximately 690 m, where the potential temperature was approximately 280 K and the RH was approximately 20%, and the extinction coefficient was also reduced to 0.7 km$^{-1}$. Due to the cooling effects of surface longwave radiation, the ground inversion layer formed from the surface at a depth of approximately 100 m. At this time, the potential temperature gradient underwent an obvious change at 600 m. The inversion intensity levels below 600 m was weaker, and the height $H_\theta$ was approximately 690 m. The most negative value of the extinction coefficient gradient appeared at approximately 500 m at this time, and the extinction coefficient below 690 m was much higher than that observed above 690 m, indicating that aerosol particles were mainly concentrated below the

inversion layer (Baumbach and Vogt, 2003) and that the $H_\theta$ calculated by radiosonde is basically consistent with $H_c$ determined by Lidar.

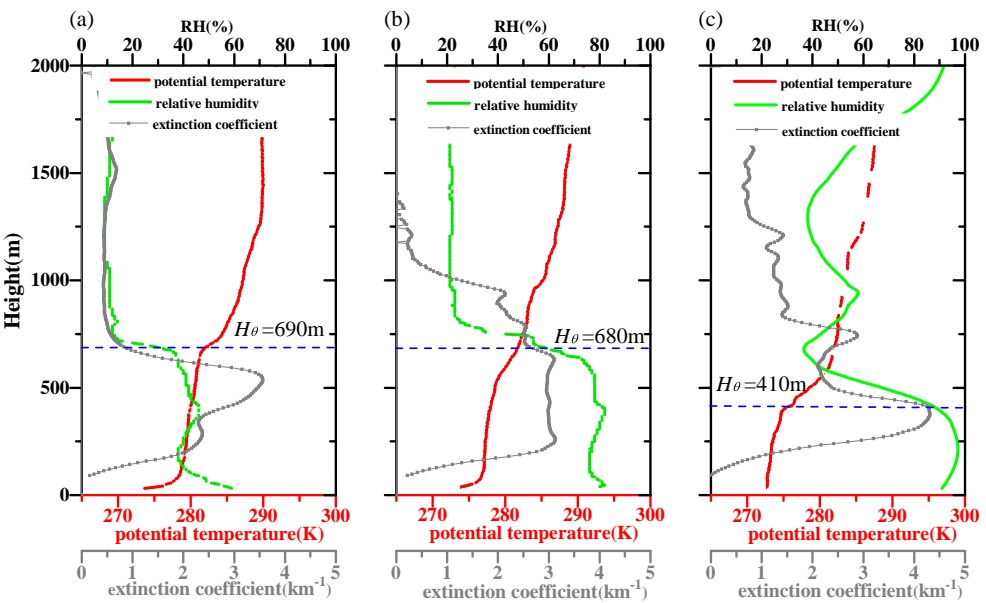

**Figure 5.** Vertical profiles measured from the ZBAA meteorological station. (a) 20:00, 16 December 2016 (b) 20:00, 17 December 2016 (c) 08:00 21 December 2016. Red line: potential temperature (K); Green line: RH (%); Grey line: extinction coefficient (km$^{-1}$); Blue dotted line: ABL height determined by radiosonde and expressed in $H_\theta$(m).

At 20:00 on 17 December, ground inversion started to form. $H_\theta$ was observed at approximately 680 m, and the potential temperature at this level was still at approximately 280 K, though the RH reached nearly 60%. Below this height, the whole atmosphere layer had developed a high humidity layer with an RH of nearly 80% from the ground, and the corresponding extinction coefficient had also increased significantly. The extinction coefficient between 250~600 m was almost 3 km$^{-1}$, revealing that the concentration of aerosols had increased significantly. Combined with the wind direction at this time (Fig.7d), it is clearly observed that the transport of easterly winds moved considerable levels of water vapor from Bohai Bay (approximately 200 kilometers east of Beijing), which promoted the hygroscopic growth of aerosol particles (Svenningsson et al., 1992; Chuang, 2003; Pan et al., 2009). At 08:00 on 21 December, the potential temperature distribution in the morning was different from that observed at 20:00, and the surface temperature had begun to increase as solar radiation was received and the $H_\theta$ was approximately 410 m. The value below 300 m was nearly 95%, while the maximum extinction coefficient, which exhibited bimodal features, reached nearly 4 km$^{-1}$. The altitude at which the extinction coefficient reached peak levels in the lower layer was approximately $H_\theta$. By means of analyzing and comparing $H_c$ and $H_\theta$ values it is apparent that when concentrations of PM$_{2.5}$ were high, the accumulation of pollutants was mainly accompanied by the inversion layer in the atmosphere. The potential temperature gradient at the inversion layer is generally larger. Even though $H_c$ reflects aerosol scattering information and $H_\theta$ denotes potential temperature characteristics, there is a strong correlation between them with a correlation coefficient

of approximately 72%. As shown in Fig.4, $H_\theta$ was significantly higher than $H_c$ determined by the three methods based on the Lidar extinction coefficient.

### 3.4  Boundary layer structure observed by WPR

The ground is the most important sink of atmospheric momentum, and the wind speed is zero at the Earth's surface. The ABL wind speed gradually changes from the Earth's surface to the geostrophic winds measured at high altitudes, and the wind information extracted from the WPR has been widely used to determine the ABL height (Cohn and Angevine, 2000; Bianco and Wilczak, 2002). A comprehensive review of estimated convective boundary layer heights is given by Seibert et al (2000). Because our WPR wind speed observations reveal the presence of many low-level jets or maxima, we used the height of the low level maximum wind speed as the ABL height (Banta, 2008; Pichugina and Banta, 2010; Devara et al., 1995), expressed in $H_u$. Figure 6 shows wind speed profiles observed by WPR on December 15, 17, 19 and 21, from which we find that wind extremes and weak jets often occur at below 1000 meters.

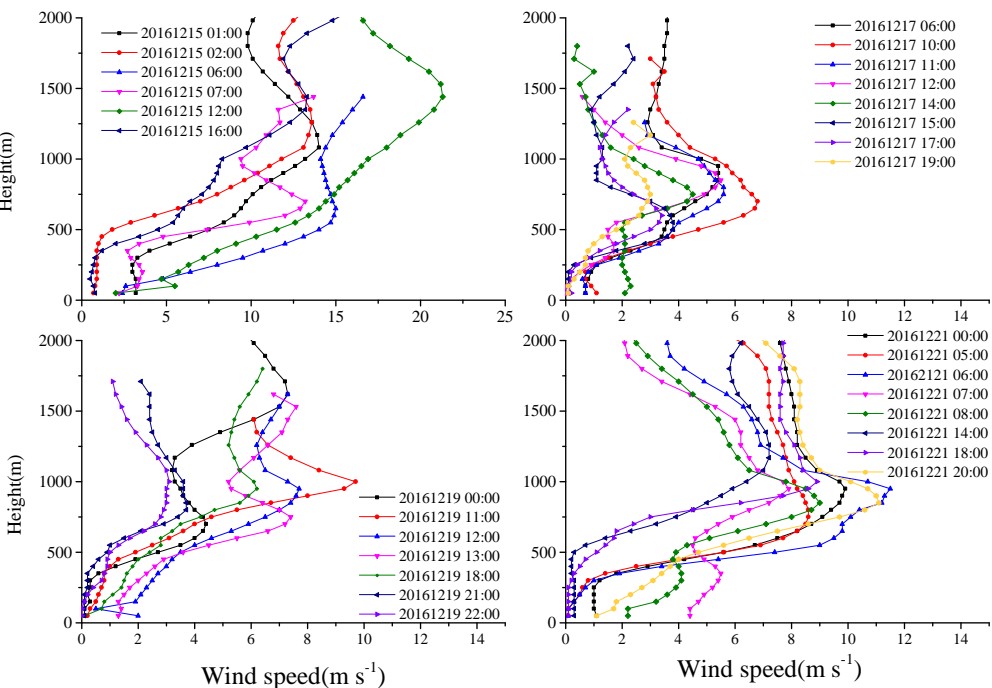

**Figure 6.** Vertical wind profiles measured by WPR on December 15, 17, 19 and 21, 2016. December 21 is the most polluted day when there are obvious low-level jets.

As shown in Fig. 4, $H_u$ was much higher than $H_c$, especially on the unpolluted day when the $H_u$ was approximately 1500 m. With an increase in PM$_{2.5}$ concentrations, $H_u$ also decreased; that is, the height of the low-level maximum wind velocity should also decrease when the atmosphere is polluted. In this case, $H_u$, $H_c$ and $H_\theta$ were very close. To analyze the influence of the boundary layer's wind structure on pollutants, we further discuss the representative wind speed and direction profiles of

the three typical days for the unpolluted and polluted conditions. As is shown in Fig. 7, wind speeds below 1000 m did not exceed 6 m s$^{-1}$ on 17 December, and the $H_u$ determined by WPR ranged from 750~1000 m. At 12:00 a typical "nose" profile distribution formed according to the wind profile with a maximum value observed in the middle of the wind profile. The wind direction profiles show that the wind direction from the ground to approximately 750 m was northeast ($0° \sim 90°$), and the wind direction observed above 750 m was northwest ($270° \sim 360°$). Furthermore, the wind directions from 750~2000 m remained

basically stable in the northwest direction, echoing geostrophic winds. At this time, the height of the maximum wind speed fit well to the height of the wind direction and geostrophic wind began to form. In addition to 12:00 on 17 December, at other four times almost strictly north-eastern winds formed below 1000 m, and winds began to transform into the northwest winds at different heights. Wind speeds increased to some extent on 21 December with typical "nose"-type wind speed distributions observed at 00:00, 08:00 and 20:00. From the ground to 2500 m steady southwest winds formed at 00:00, and the maximum

wind speed was approximately 900 m. At this time, the extinction coefficient was also very low above 750 m (shown in Fig. 6), demonstrating that pollutants also formed below a height of $H_u$. At 12:00 the wind speed began to decrease slightly from the ground, and no obvious changes were observed beyond approximately 1100 m where wind speeds reached approximately 4 m s$^{-1}$. Except at 00:00 on 21 December when southwest winds prevailed from the ground to high altitudes, the wind directions shifted to the northwest at other times though these wind speeds were less strong below 500 m, and wind speed maximum

values formed at a height of approximately 1000 m.

     Wind directions observed on 22 December were northwest from the low layer to the high layer, but the distribution of wind velocity profiles differed from that of 21 December. Wind speeds were based on no significant maximum value area, and the maximum wind speed of 500 m approached close to 12 m s$^{-1}$. According to the extinction coefficient distribution (shown in Fig.4), the PM$_{2.5}$ concentration was greatly reduced on this day. $H_u$ determined by WPR and $H_\theta$ obtained by radiosonde were

relatively similar at this time, and both were far higher than $H_c$.

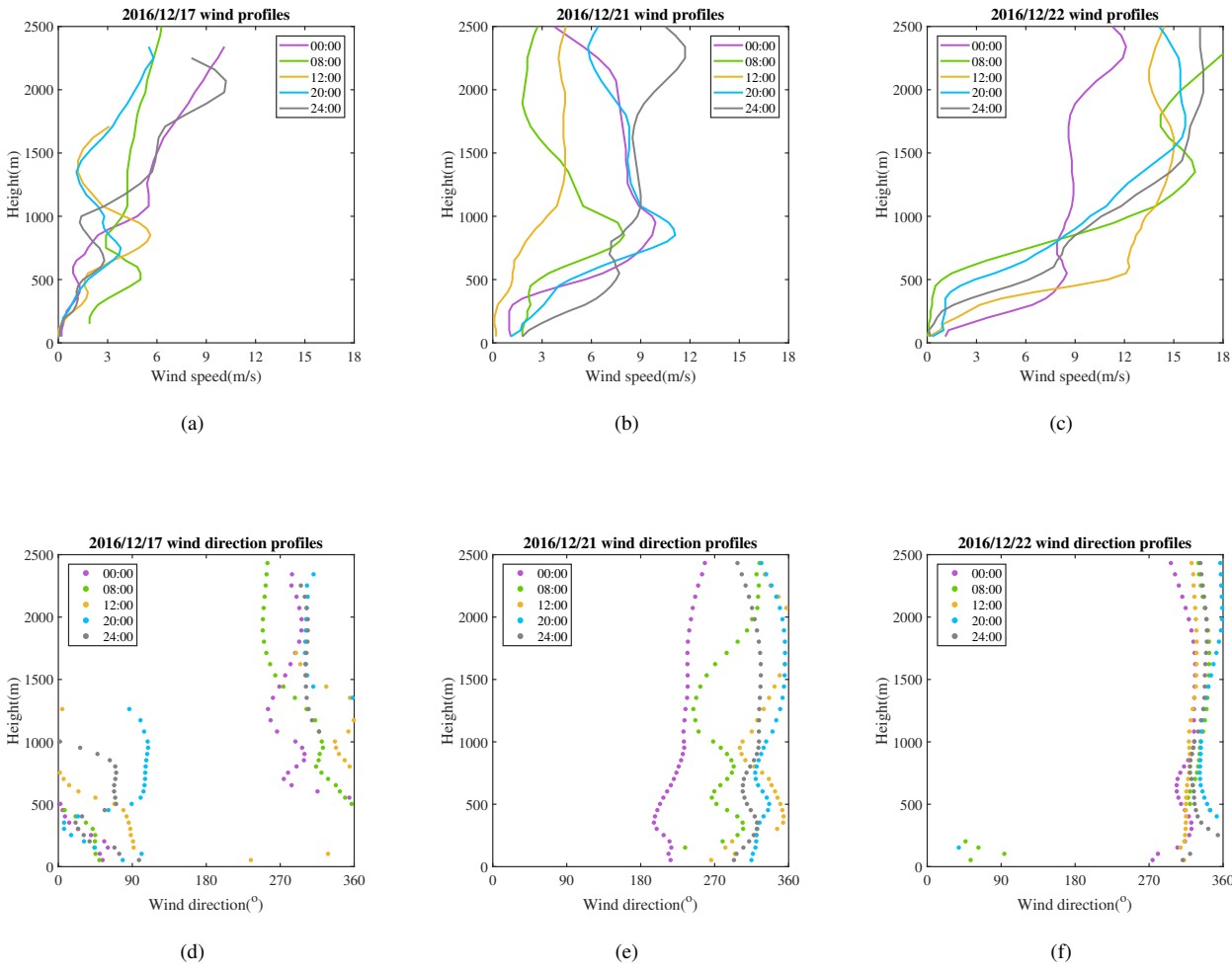

**Figure 7.** Vertical profiles of wind speed (m s$^{-1}$) and wind direction ($^\circ$) observed by WPR. (a) and (d): 16 December, 2016; (b) and (e): 21 December, 2016; (c) and (f): 22 December, 2016.

### 3.5 Boundary structure and turbulence quantities observed from the 325 m tower

Based on high-resolution gradient observations (15-layer mean and 7-layer turbulence measurements), we can analyze the relationship between PM$_{2.5}$ and low-level turbulence, average wind speed and temperature. As shown in Fig. 8, both turbulent kinetic energy (TKE) and friction velocity (u$_*$) at 140 m and 280 m were inversely correlated with ground level PM$_{2.5}$ concentrations. TKE and u$_*$ can be calculated as follows (Stull, 1988): $TKE = \frac{1}{2}(\overline{u'^2}+\overline{v'^2}+\overline{w'^2})$; $u_* = (\overline{u'w'}^2 + \overline{v'w'}^2)^{1/4}$. The maximum TKE levels observed on unpolluted days (15 to 16 December) reached approximately 7 m$^2$ s$^{-2}$ at 140 m while the TKE levels measured on hazy days (17 to 21 December) decreased sharply to very low values. After the start of the period of heavy haze pollution, TKE levels remained relatively low, and the change in TKE was not as significant when concentrations of PM$_{2.5}$ increased from 200∼400 $\mu$g m$^{-3}$. On the other hand, the time series for u$_*$ is slightly different from that of TKE. It

seems that the inverse correlation between u$_*$ and PM$_{2.5}$ is more obvious than that of TKE for the heavy pollution period. In fact, even during the period of heavy haze, a slight fluctuation (diurnal variation) in PM$_{2.5}$ concentrations can be observed, and the diurnal variation in u$_*$ follows the opposite pattern to that of the PM$_{2.5}$ phase.

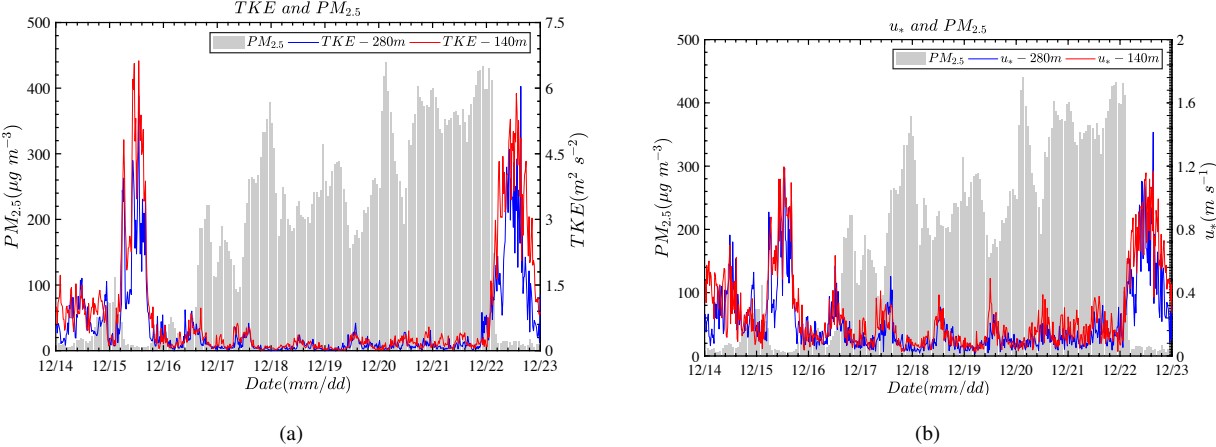

(a)                                                                      (b)

**Figure 8.** Time series of (a) turbulent kinetic energy (m$^2$ s$^{-2}$) and (b) friction velocity (u$_*$, m s$^{-1}$) at 140 m (red line) and 280 m (green line); PM$_{2.5}$ ($\mu$g m$^{-3}$) concentrations are also shown in the figures (grey column).

To understand the vertical structure characteristics of the ABL observed from the tower during the polluted and unpolluted periods, profiles for wind, potential temperature ($\theta$), TKE and sensible heat flux ($\overline{w'\theta'_v}$) for the lower boundary layer are also given. At night (see Fig.9), the wind speed profile for the clear day basically follows a logarithmic distribution and the potential temperature changes little from the ground to approximately 300 m. For turbulent values, TKE gradually decreased from approximately70 m, and sensible heat flux was basically negative. On the polluted day, wind speeds from the lower layer to the upper layer were valued at less than 2m s$^{-1}$. At this time, the change in potential temperature was not very large from the ground to approximately200 m, indicating that the atmosphere basically maintained neutral levels of stratification. A pronounced inversion layer cap is also observed to approximately200 m from the surface. TKE levels were basically maintained at close to zero. Note that at this time the sensible heat flux measured at above 80 m remained at close to zero. Close to the ground, sensible heat flux was slightly positive, demonstrating that when pollution occurred and especially when the inversion layer existed, heat flux transport was suppressed. At night, surface longwave radiative cooling was restrained to a certain extent, and the weakening of turbulence activities aggravated pollution levels once again.

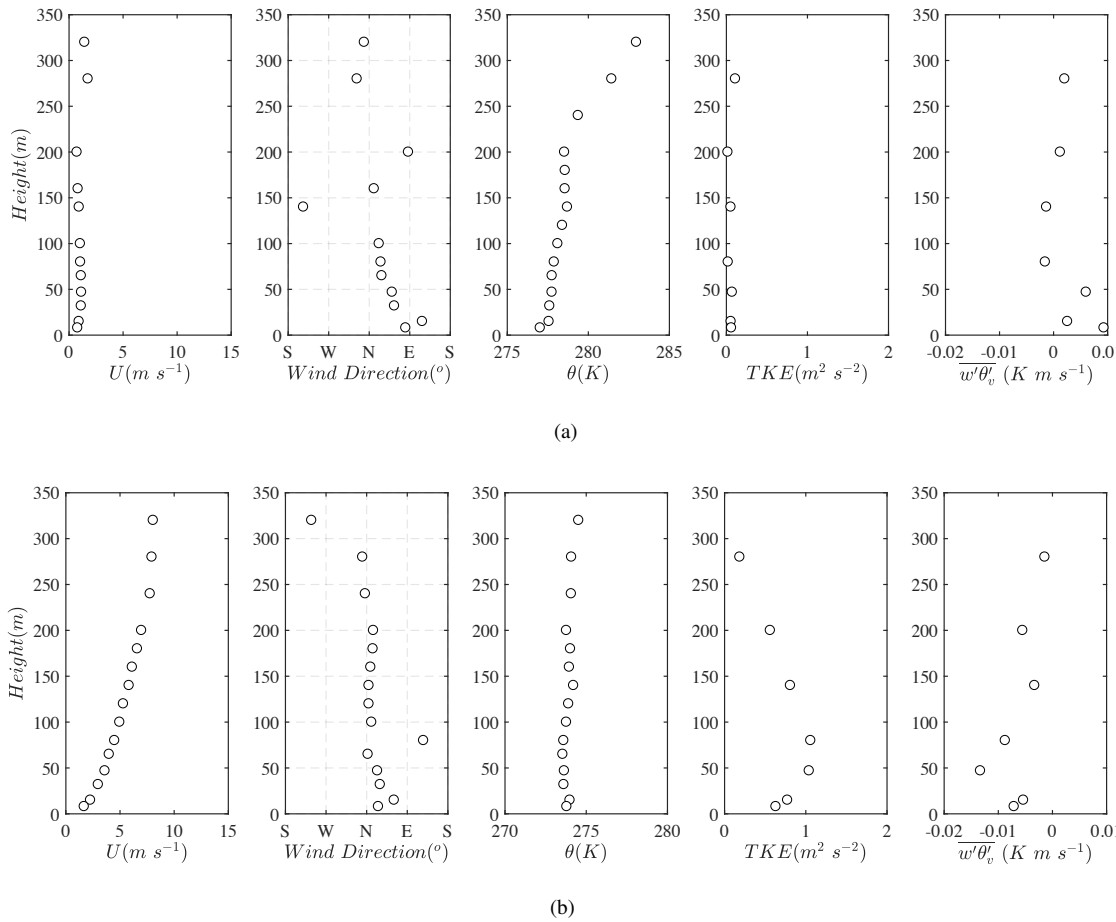

**Figure 9.** Vertical profiles of wind speed (U), wind direction (WD), potential temperature ($\theta$), turbulent kinetic energy (TKE) and sensible heat flux $(\overline{w'\theta'_v})$ at 23:20 on 19 December, 2016 (a), 22 December, 2016 (b).

In the daytime on 21 December 2016 (see Fig. 10), the PM$_{2.5}$ concentrations reached roughly 400 $\mu$g m$^{-3}$, wind speeds were low, TKE remained zero value, and the potential temperature observed from the tower during the period of pollution basically denoted neutral stratification. The sensible heat flux was positive, but the value was basically measured as 0.02 K m s$^{-1}$. At noon on 22 December, when the weather had improved, wind speeds were clearly higher. TKE still reached a maximum at 47 m. The influence of the urban canopy was stronger at heights of below 47 m. Unlike on the polluted day, levels of sensible heat flux were higher at this time, and the lower layer reached a value of 0.1 K m s$^{-1}$. The tower observation data clearly show that levels of sensible heat flux decreased significantly in the daytime during the haze episode because of the higher levels of solar radiation scattering by particles.

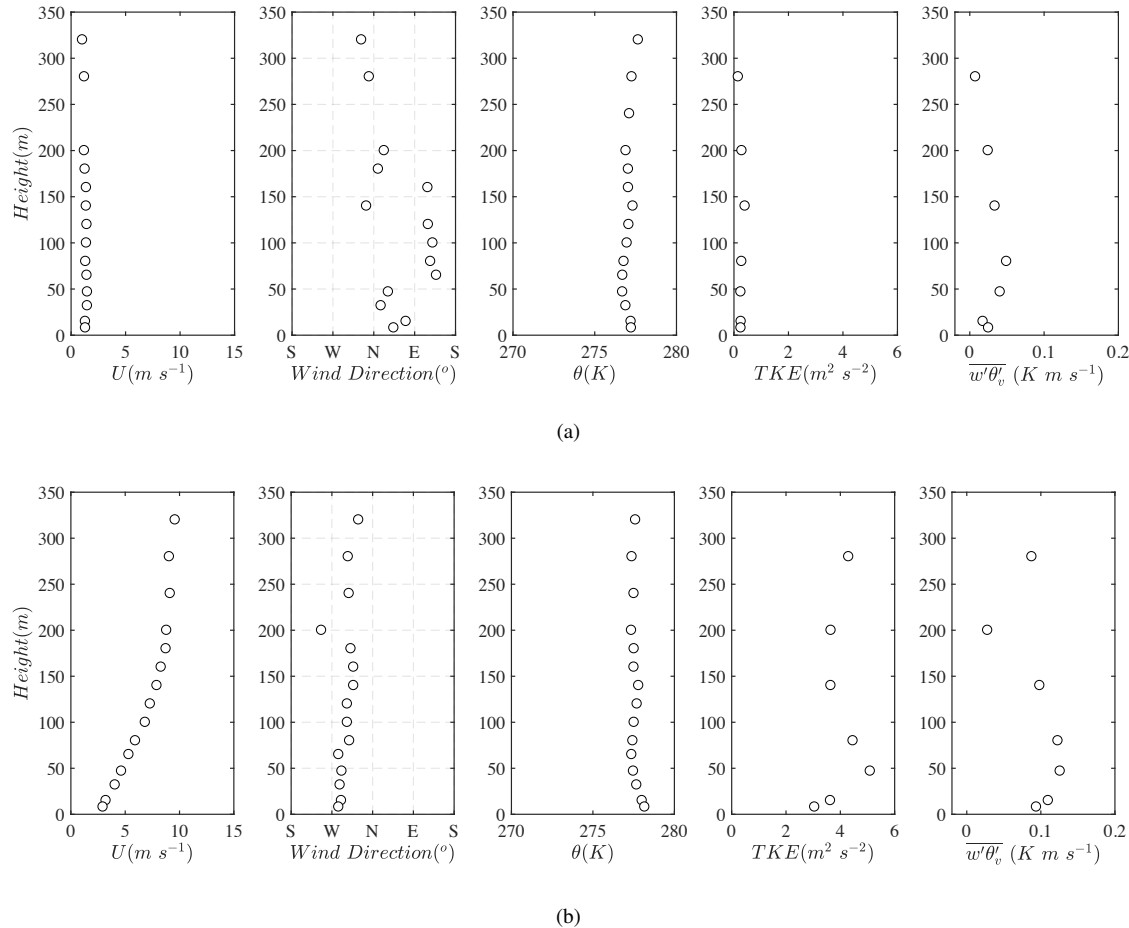

**Figure 10.** Vertical profiles of wind speed (U), wind direction (WD), potential temperature ($\theta$), turbulent kinetic energy (TKE) and sensible heat flux ($\overline{w'\theta'_v}$) measured at 12:00 on 21 December, 2016 (a) and 22 December, 2016 (b).

According to the concentration of PM$_{2.5}$ observed, the haze pollution episode can be divided into different grades. Statistical mean values of surface visibility (Vis), wind speed (U), RH, ABL height and turbulent fluctuations are also calculated. As is shown in Table2, the statistical averages further confirm the conclusions of the previous analysis. Rather, when the concentration of PM$_{2.5}$ is high, visibility and wind speed decrease while RH increases significantly. Our results show that due to the
5   accumulation of aerosol particles, $H_c$ is heightened slightly while $H_u$ declines by roughly 300m. The Lidar results overestimate the ABL height at night (Quan et al., 2013). In this study the ABL height also increased due to the accumulation of pollutants during the period of heavy pollution. The turbulence levels observed also exhibit a decreasing trend occurring during the haze pollution episode, further demonstrating that turbulent activities are inhibited to a certain extent. We note that there appears to be only slight differences between the ABL height under slightly, moderately, heavily, and seriously polluted conditions.
10  Since the data examined in this study apply to only one period of heavy pollution, the minor differences observed may not be statistically different across categories except when compared to "good" air quality conditions.

**Table 2.** Averaged values of visibility (Vis), wind speed (U), relative humidity (RH), ABL height ($H_u$, $H_c$), turbulent kinetic energy (TKE), friction velocity ($u_*$), momentum flux $(\overline{u'w'})$, and sensible heat flux $(\overline{w'\theta_v'})$ for different levels of pollution. Corresponding relationships between air pollution levels and PM$_{2.5}$ concentrations are as follows: good ($0\sim75\mu$g m$^{-3}$), slightly polluted ($75\sim115\mu$g m$^{-3}$), moderately polluted ($115\sim150\mu$g m$^{-3}$), heavily polluted ($150\sim250\mu$g m$^{-3}$), and seriously polluted ($>250\mu$g m$^{-3}$).

| Quality Level | Vis (km) | U (m s$^{-1}$) | RH (%) | TKE (m$^2$ s$^{-2}$) | $u_*$ (m s$^{-1}$) | $\overline{u'w'}$ (m$^2$ s$^{-2}$) | $\overline{w'\theta_v'}$ (K m s$^{-1}$) | $H_u$ (m) | $H_c$ (m) |
|---|---|---|---|---|---|---|---|---|---|
| Good | 9.9 | 4.0 | 39 | 1.23 | 0.31 | 0.08 | 0.0115 | 1124 | 358 |
| Slightly | 5.8 | 1.5 | 73 | 0.22 | 0.15 | 0.02 | 0.0022 | 837 | 484 |
| Moderately | 6.7 | 1.3 | 64 | 0.51 | 0.2 | 0.03 | 0.0032 | 750 | 502 |
| Heavily | 4.6 | 1.67 | 63 | 0.16 | 0.12 | 0.00045 | 0.0078 | 873 | 510 |
| Seriously | 2.0 | 1.38 | 81 | 0.15 | 0.11 | 0.0057 | 0.0038 | 844 | 518 |

## 4 Conclusions

In this paper a red warning haze pollution period running from 14 to 22 December 2016 occurring in Beijing was studied using various observational techniques. Atmospheric boundary layer structures and turbulence characteristics are the focus of this paper. Observational techniques used include not only remote sensing techniques, e.g., Lidar and WPR, but also direct measurement techniques, e.g., ground-based radiosonde technologies and the 325m meteorological tower. Our research results show that during the studied period of heavy haze pollution, the Beijing area was controlled by a stagnant weather system. Water vapor transport levels has increased relative humidity levels at below 600m, greatly promoting the hygroscopic growth of $PM_{2.5}$. The height of the ABL observed by Lidar ($H_c$) was measured at roughly 500∼750m. The inversion layer is closely related to the concentration of pollutants. Pollutants emitted in the ABL generally accumulate under the inversion layer. The inversion layer's height decreased significantly during the pollution period, and the lowest value was measured at below 500m. The present study shows that $H_c$ did not seem to decline during the heavy pollution episode due to the accumulation of pollutants. According to the potential temperature gradient method, the ABL height calculated by radiosonde ($H_\theta$) is in good agreement with $H_c$ with a correlation coefficient of close to 72%. The ABL height ($H_u$) determined by WPR is higher than that of $H_c$, and $H_u$ clearly decreased when heavy pollution levels of closer to $H_c$ and $H_\theta$ occurred. Low TKE, $u_*$ and $PM_{2.5}$ values were observed to be inversely related according to the tower data. Turbulent fluxes varied very little with altitude, but the sensible heat flux measured at night was slightly positive close to the Earth's surface, indicating that the cooling effect is inhibited by long-wave radiation from the ground. Due to higher levels of solar radiation scattering by particles, sensible heat flux in the daytime was greatly reduced. Consequently, the suppression of turbulence leads to more serious pollution outcomes.

Although different boundary layer heights can be obtained using various techniques, it seems that the ABL's height measured by Lidar can better reflect pollution levels that accumulate during periods of heavy haze pollution, and the ABL's height measured by radiosonde is also in good agreement with the $H_c$ measured by Lidar, which is useful for the study of the atmospheric pollution boundary layer based on conventional observations. Our research importantly found that the ABL's height measured by WPR ($H_u$) is high. However, as the definition of ABL's height determined by different means varies within this class, they present respective roles and levels of significance. In our future work we will strive to parameterize the relationship between friction speed and $PM_{2.5}$ concentrations, as they exhibit strong statistical correlations (negative correlations) for use in numerical models of air pollution. In addition, it will be meaningful to explore the correlations between dynamic, thermal and material (concentration) boundary layer heights (expressed by $H_u$, $H_\theta$ and $H_c$, respectively) through more observations.

*Competing interests.* All the authors have declared that no competing interests exist

*Acknowledgements.* The authors thank Dr. Aiguo Li from the Institute of Atmospheric Physics of the Chinese Academy of Sciences for his assistance with the use of 325 m tower data. This work was supported by the National Key Research and Development Program of China (2017YFC0209605) and the National Natural Science Foundation of China (Grant No. 11472272).

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
