# Peer review of "Multiple technical observations of the atmospheric boundary layer structure of a red warning haze episode in Beijing"

_Atmospheric Measurement Techniques, 2018_

## Referee Comment (RC1) · Anonymous Referee #2 · 27 Feb 2019

Within this study, the authors discuss a weeklong period when there was extremely poor air quality in Beijing in December of 2016. The authors use data from a 325-m tower, a backscatter lidar, 12-hourly radiosondes, and a wind profiling radar to analyze the atmospheric boundary-layer structure during this time frame. Boundary-layer heights are determined from the lidar, wind profiler, and radiosonde, which are intercompared and used to evaluate how the height changes depending on the air quality. The authors also find that the TKE and $u\_*$ are inversely correlated with PM2.5 concentrations.

Unfortunately, there are numerous major concerns with this study which are listed be-

low. Furthermore, there is little if any new material in this paper. The authors are using well-established techniques to determine the PBL height, and some are improperly applied. One key finding is that the aerosols hygroscopically grew as water vapor increased, but this has been documented in many previous studies (e.g., Svenningsson et al 1992, Chuang 2003, Pan et al 2009). Additional key findings are that the PBL heights were lower during polluted days and that turbulent activity is inhibited on these days, which have also been thoroughly documented (e.g., Schäfer et al 2006, Quon et al. 2012, Petäjä et al 2016) The material in the paper may appear novel since the authors do not relate their results and findings to previous studies in their paper. Given the numerous concerns with the manuscript listed below and the aforementioned limited novelty of the study, I unfortunately cannot recommend this paper be acceptable for AMT.

Major comments:

a) In Sect. 3.3, the authors use the maximum gradient in the potential temperature to identify the PBLH in radiosonde data. This method is only appropriate in the convective PBL (as in the cited Hennemuth and Lammert, 2006 study). Based on the profiles shown in Fig. 4, the PBL was stable and this method is inappropriate to determine the PBLH. Seidel et al (2010) further discusses how the PBLH should be determined in a SBL and the problems of using the maximum gradient in potential temperature in these conditions. It's also important that the authors clarify what they mean by the PBL height. Is the layer that is currently being mixed as it interacts with the surface the desired layer? This is often very shallow in the SBL. Alternatively, do the authors mean the height of the residual layer (from the previous day's CBL)? If the later is the case, then the potential temperature method may be appropriate. However, it must be noted that this height (top of residual layer) is not where pollutants are actively mixed, which is the desired quantity for air pollution studies. See also Keller et al (2011) for a method in which the SBL height can be calculated from a temperature profile.

b) In Sect. 3.4, the authors identify the PBLH as the maximum in the wind speed. While

this method has been used before (as the authors note in the cited studies), it is only appropriate for identifying the PBLH when a low-level jet is present, which there is not during much of this study. Instead, the authors should use a more appropriate method for determining the PBLH from a WPR, such as Cohn and Angevine (2000) or Bianco et al (2002). It important to consider that WPR are often currently unable to measure the PBL height in the stable boundary layer due to the PBL being shallow (below the minimum range gate) and there is rarely a refractive index maximum signature. If a different more appropriate method is not used, then the WPR PBL height measurements should be removed from this study.

c) This study could benefit from a discussion of the synoptic and mesoscale meteorology. Based on Fig. 2, it looks like there were cold fronts (strong NW winds) on both 12/15 and 12/22 which advected pollutants away and resulted in air quality. Between the fronts, the PM2.5 slowly increased as the air was stagnant (weak and variables winds in between) and pollutants emitted locally likely slowly built up. The wind direction also seems to be cyclical every day, perhaps this is in response to a local mountain-valley circulation around Beijing? This warrants more investigation. It appears that these factors, namely the stagnant air, dominate over the PBL processes in resulting in the poor air quality.

d) While the authors provide numerous references early in the manuscript, there is little-to-no discussion of how the key findings and results compare to previous studies. For example, how does this haze episode compare to other poor air quality case studies in Beijing and other areas, considering both meteorology and boundary-layer processes? The authors should relate the hygroscopic growth to that observed in other studies (there are many). There have also been many other studies comparing lidar PBL heights with those from radiosonde and/or wind profilers, mainly focused on the convective boundary-layer where the top of the boundary-layer is clearly defined. These studies should be related to as well.

Minor comments:

a) P. 1 line 13: 'Tubrulent activities were great inhibited during haze pollution'. This must be rephrased, as the evidence in the paper does not support this statement. It is unclear if the haze suppresses turbulence, or if weak turbulence results in poorer air quality.

b) P. 2, line 8: Please define all the acronyms, such as for URBAN, MIAGE, and SURF (similarly to how COST was defined and spelled out).

c) Figure 1: Add a reference scale for distance on the plot. Currently, this map is not very informative in itself. It would be beneficial to add other details to the map relevant to the study (i.e., other important locations, elevation map, any significant pollutant sources, etc) if possible.

d) P. 3 line 14: It is stated there are 15 platforms on the tower, but only 14 levels are subsequently listed. This inconsistency should be rectified.

e) P. 4 line 16: List out the six air pollutants measured.

f) Table 1: What is the difference between 'heavy polluted' and 'serious polluted'? Describe in the text these categories (and all the other ones). It is unclear which category is worse. These are later defined in Table 2 at the end of the manuscript, but these definitions should be moved to the discussion of table 1.

g) P. 5, line 3: Here, the authors claim that the lack of diurnal variation of temperature and RH is due to heavy pollution. This is plausible, but another reason is more likely. The high RH and low visibility, secondarily combined with the low wind speed, all indicate that fog or low stratus is present. The fog or stratus itself would greatly reduce insolation and daytime heating. This would effect would likely dominate over aerosol effects in reducing solar radiation reaching the surface. The discussion here needs to be modified accordingly. It would also be useful to include a plot of cloud cover somewhere.

h) P. 5 lines 6-7: RH cannot be used to quantify the total amount of water vapor in the

air, as RH is highly-dependent on the temperature. The increase in moisture discussed here may simply be due to a lower temperature, not an increase in water vapor content. Please use mixing ratio or another conserved quantity.

i) Sect 3.2: Rename this section "Lidar observed boundary layer heights", as the WPR and radiosonde and not discussed in it. These paragraph needs to be rewritten and better organized. Start by explaining why the extinction profile can be used to identify the ABL height, then lead into three different methods that can be used that will yield different estimates, but also each have their own limitations. Describe each method separately, as currently the description of all three is spread throughout the entire section.

j) P. 7, line 10: What is the dilation of the Haar wavelet? The determined ABL height has been shown to be sensitive to this in many previous studies.

k) Figures 3 and 4: Why is extension near the surface so small? I would expect the extinction profile to be nearly uniform (most of the time, except in the presence of clouds) throughout the entire PBL. This leads me to believe that the overlap correction for the lidar is not correctly determined and/or applied. Also, in calculating the extinction coefficient, is attenuation considered? Also, please state if time here (and throughout the manuscript) is local time or GMT.

l) P. 8 line 8: What is meant by 'transformation zone'? This is not a commonly used term. m) P. 8 line 10: This reasoning is insufficient to discount the lidar observations, especially given following concerns with the PBLH determined from both the radiosonde and wind profiler. If the extinction is plotted on a logarithmic scale (which is more appropriate given the large dynamic range that it can vary), a real gradient associated with the PBLH determined by the lidar would likely be more apparent. Hints of this gradient are apparent on the plot now during these clean periods at around 250 m on 12/15 and 12/22.

n) P. 8, line 23: The fact that the PBL was statically stable is unsurprising given the

timing of the radiosonde profiles at 08:00 and 20:00 local time, which are respectively about 30 min after sunrise and 3 hours after sunset during the experimental period. Thus, a nocturnal inversion would barely erode by 08:00 (if at all, depending on the energy balance as insolation is small) and a nocturnal inversion would have formed by 20:00. Due to the timing, these profiles are not representative of conditions during the day when pollutants are actively mixed. I suspect that if the profiles were during the midday (noon local time or early afternoon) the profiles would indicate instability and mixing. This must be discussed at the very least.

o) P. 8, line 28-29: Based on this description (and the profiles), it seems as though the top of the stable boundary layer where mixing is occurring is at 100 m (not 700 m), while the top of the decoupled residual layer is at 700 m.

p) P.9, line 5: Why would an easterly wind advect water vapor? This is unclear and not supported by the data. A map of the water vapor field at the time would be needed to support this.

q) P. 9 line 6: The authors should provide a reference here for hygroscopic growth, as it has been detailed in numerous prior studies.

r) P. 9 line 8: The profile still appears stable in panel c) as the inversion is not completely eroded yet with potential temperature increasing from the surface upwards, thus the ground inversion is still apparent (although weaker) than in panels a) and b).

s) P. 9 line 11: Why is $H\_theta$ at 604m? By eye, the maximum in the gradient of the potential temperature (and humidity) is near 350 m.

t) Figure 4: The tick marks on the right-most y-axis in panel c) are not aligned with all the other tick marks, which is confusing and deceptive. Also, why do all the profiles start at around 50 m? Is this height above ground level or height above sea level?

u) P. 9 line 13: Rephrase the sentence "At the inversion layer, it was easier to appear the larger value of the potential temperature gradient", as its meaning is unclear.

v) P. 9 line 16: Again, please provide evidence to support this statement. Perhaps in Fig. 4 a few panels could be added (and a supporting discussion in the text) that show profiles where there is no inversion layer apparent (or it is weak).

w) P. 11 line 4: The equation for u_* is incorrect. The u'w' and v'w' quantities should both be squared. Make sure it is calculated correctly in Fig. 6, b as well.

x) Fig. 7: Please make the colorbar for the wind direction (in leftmost panel) circular. With the current colorscale, wind direction at 359 deg is red and 1 deg is blue, even though there's little difference in the wind direction. It would also be beneficial to provide plots from daytime conditions when the PBL is well-mixed during the pollution episode, as both of the selected time periods are near midnight.

y) Table 2 (and its discussion): There appears to be little difference between slightly, moderately, heavily, and seriously polluted conditions. There is no clear trend in the listed variables with increasing pollution. I suspect that the calculated small differences are not statistically different between categories, except compared with 'good' air quality. This is unsurprising, given the meteorology remains roughly constant during all periods between times when the air quality is 'good' when pollutants slowly build up. This should be discussed.

z) P. 15 line 6: Again, there is no evidence in the manuscript that north-easterly winds brought pollutants and water vapor to Beijing. Based on Fig. 2, it looks like this first increase in pollution was the first night when winds were light, thus pollutants likely emitted locally were not dispersed and built up quickly in a shallow layer near the surface.

aa) P. 15 line 13: Rephrase the sentence 'The turbulent fluxes . . . radiation from the ground", as its meaning is unclear.

bb) P. 15 line 15: How is turbulence suppressed when the sensible heat flux is positive? Normally a negative heat flux with cooling near the surface will strengthen the stability

limiting turbulence, while a positive heat flux weakens stability.

Editorial corrections:

a) The manuscript needs heavy editorial corrections thoughout.

b) P. 3, line 12: 49 m above sea level is listed twice in this sentence. Remove one instance.

c) P. 3, line 13: Why is 'August 1979' here?

References:

Bianco, L., Wilczak, J.M., and White, A. B., 2002. Convective boundary layer depth: Improved measurement by Doppler radar wind profiler using fuzzy logic methods. Journal of Atmospheric and Oceanic Technology, 19(11), pp.1745-1758.

Chuang, P.Y., 2003. Measurement of the timescale of hygroscopic growth for atmospheric aerosols. Journal of Geophysical Research: Atmospheres, 108(D9).

Cohn, S.A. and Angevine, W.M., 2000. Boundary layer height and entrainment zone thickness measured by lidars and wind-profiling radars. Journal of Applied Meteorology, 39(8), pp.1233-1247.

Keller, C.A., Huwald, H., Vollmer, M.K., Wenger, A., Hill, M., Parlange, M.B. and Reimann, S., 2011. Fiber optic distributed temperature sensing for the determination of the nocturnal atmospheric boundary layer height. Atmospheric Measurement Techniques, 4.

Pan, X.L., Yan, P., Tang, J., Ma, J.Z., Wang, Z.F., Gbaguidi, A. and Sun, Y.L., 2009. Observational study of influence of aerosol hygroscopic growth on scattering coefficient over rural area near Beijing mega-city. Atmospheric Chemistry and Physics, 9(19), pp.7519-7530.

Petäjä, T., Järvi, L., Kerminen, V.M., Ding, A.J., Sun, J.N., Nie, W., Kujansuu, J.,
Interactive
comment

Virkkula, A., Yang, X., Fu, C.B. and Zilitinkevich, S., 2016. Enhanced air pollution via aerosol-boundary layer feedback in China. Scientific Reports, 6.

Quan, J., Gao, Y., Zhang, Q., Tie, X., Cao, J., Han, S., Meng, J., Chen, P. and Zhao, D., 2012. Evolution of planetary boundary layer under different weather conditions, and its impact on aerosol concentrations.

Schäfer, K., Emeis, S., Hoffmann, H. and Jahn, C., 2006. Influence of mixing layer height upon air pollution in urban and sub-urban areas. Meteorologische Zeitschrift, 15(6), pp.647-658.

Seidel, D.J., Ao, C.O. and Li, K., 2010. Estimating climatological planetary boundary layer heights from radiosonde observations: Comparison of methods and uncertainty analysis. Journal of Geophysical Research: Atmospheres, 115(D16).

Svenningsson, I.B., Hansson, H.C., Wiedensohler, A., Ogren, J.A., Noone, K.J. and Hallberg, A., 1992. Hygroscopic growth of aerosol particles in the Po Valley. Tellus B, 44(5), pp.556-569.
* * *

---

## Author Comment (AC1) · 21 Mar 2019

Dear reviewer,

Thank you very much for your comment on our paper "M", we think it is really very valuable and it helps us a lot.

In particular, as the first author, I am currently a Ph.D. student. I have really learned a lot of boundary layer knowledge from your comments. I would like to express my respect and heartfelt thanks to you !

Based on your review comments, we have added some calculations, redrawn some

of the figuers, and responded to your comments one by one in the attachment, and adopted all your suggestions. Specifically:

1) Complement all recommended references. 2) Added MODIS satellite cloud image. 3) Corrected the representation of the height of the radiosonde detection boundary layer. As you said, the maximum temperature gradient method we obtained should be the residual layer top height. 4) Recalculating and drawing with high-resolution radiosonde, giving the wind profile of the low-level jet during heavy pollution. As you pointed out, we should use the jet height (or low-level wind extreme) as the boundary layer height. The value of the boundary height has also been corrected. The drawing is shown in Figure 6. 5) Increased the daily variation of the surface wind field in Beijing and the discussion of the removal of pollutants by cold air. 6) Respond to all major and minor comments you have given.

We sent the revised paper as an attachment, and we are continuing to modify it.

Thank you very much !

Please also note the supplement to this comment:
https://www.atmos-meas-tech-discuss.net/amt-2018-391/amt-2018-391-AC1-supplement.zip

―――――――――――――――

---

## Referee Comment (RC2) · Anonymous Referee #3 · 26 Mar 2019

The authors have addressed my comments satisfactorily. I think the manuscript is acceptable for the publication.

---

## Referee Comment (RC3) · Guenter Baumbach (Referee) · 29 Mar 2019

I agree for publication of this article. It shows an overview about the different methods to determine the stability of the Atmosheric Boundary Layer and how the stability is influencing the concentration of Air pollution, indicated as PM2.5 concentration. Only in the beginning, a definition of the ABL and the different layers should be helpful, e.g. Mixing Layer, Surface Inversion Layer, Capping Inversion, Convective Mixed Layer, Residual Layer.

Usually the ABL is the turbuelnt layer with winds influenced by the earth surface. Within this definition the Wind Profile Radar (WPR – the full word should not only be used in

the abstract) gives the best result of the ABL height.

In the air quality community the mixing layer and inversion layers are more common to use in the context of air pollution concentrations. These terms are explained at the beginning of chapter 3.2. But this could be done some more concisely., perhaps extracting one day of Fig. 3 and explaining it.

In Fig. 3 the heights Hc and Hu should be marked which are reffered to later.

Table 2:The first column shows the air quality, I think these are PM2.5 concentration ranges. It should be mentioned. There is a general influence of humidity. But not so clear. That is why the humidity depends on the origin of the advected air (more wet or more dry). Important is the decrease jump at the inversion layer, shown in Fig. 4 and not the abolute value of the humidity. Page 13 line 5: . . . "Hc is even heightened slightly but Hu (not Hc) reduces by . . ."

In the conclusion the role of the mixing layer and the inversion layers should be higlighted in the context with air pollution concentrations. And these layers are best determined by radiosonde soundings and lidar. The tower measurements are also helpful to determine the surface inversions height, if the inversion layer is lower than the tower height. With the Wind Profile Radar (WPR) the height of the Atmospheric Boundary Layer (turbulent) can be well determined. But for the air pollutant concentrations (PM 2.5) the inversion Layers and the Mixing Layer are relevant which is shown in this paper.

---

## Author Comment (AC3) · 15 Apr 2019

Thank you very much for your comments on our paper! Your review and comments are very helpful to our paper, and we have learned a lot from them. We replied to your comments one by one, and sent them to you as an attachment with the revised paper.

Please also note the supplement to this comment: https://www.atmos-meas-tech-discuss.net/amt-2018-391/amt-2018-391-AC3-supplement.zip

---

## Author Response (AR1)

**Final Response Form**

Dear Editor,

We have responded to the interactive discussion of RC1, RC2 and RC3 one by one,and the response to the editorial corrections.

The following is the final form of the reply.

Thank you much for your kind help!

Fei Hu
Yu Shi

**1. Response to RC 1**

Dear reviewer,

Thank you very much for your comment on our paper, we think it is really very valuable and it helps us a lot.

In particular, as the first author, I am currently a Ph.D. student. I have really learned a lot of boundary layer knowledge from your comments. I would like to express my respect and heartfelt thanks to you!

Based on your review comments, we have added some calculations, redrawn some of the figures, and responded to your comments one by one in the attachment, and adopted all your suggestions. Specifically:

1) Complement all recommended references.
2) Added MODIS satellite cloud image.
3) Corrected the representation of the height of the radiosonde detection boundary

layer. As you said, the maximum temperature gradient method we obtained should be the residual layer top height.

4) Recalculating and drawing with high-resolution radiosonde, giving the wind profile of the low-level jet during heavy pollution. As you pointed out, we should use the jet height (or low-level wind extreme) as the boundary layer height. The value of the boundary height has also been corrected. The drawing is shown in Figure 6.

5) Increased the daily variation of the surface wind field in Beijing and the discussion of the removal of pollutants by cold air.

6) Respond to all major and minor comments you have given.

7) Carefully revised and edited the full text of the paper, and many details were rewritten.

Thank you very much !

Yours sincerely,

Yu Shi (first author)

Fei Hu (corresponding author)

● **Response to the Major comments**

**Major comments (a)** : In Sect. 3.3, the authors use the maximum gradient in the potential temperature to identify the PBLH in radiosonde data. This method is only appropriate in the convective PBL (as in the cited Hennemuth and Lammert, 2006 study). Based on the profiles shown in Fig. 4, the PBL was stable and this method is inappropriate to determine the PBLH. Seidel et al (2010) further discusses how the PBLH should be determined in a SBL and the problems of using the maximum gradient in potential temperature in these conditions. It's also important that the authors clarify what they mean by the PBL height. Is the layer that is currently being mixed as it interacts with the surface the desired layer? This is often very shallow in the SBL. Alternatively, do the authors mean the height of the residual layer (from the previous day's CBL)? If the later is the case, then the potential temperature method may be appropriate. However, it must be noted that this height (top of residual layer) is not where pollutants are actively mixed, which is the desired quantity for air pollution studies. See also Keller et al (2011) for a method in which the SBL height can be calculated from a temperature profile.

**Response:** The maximum temperature gradient method we used is indeed inaccurate and is affected by the depth of residual layer. In the revised paper, we discussed and revised this point and gave relevant references. On the other hand, the radiosonde data we used was obtained at 08:00 and 20:00 local time, and they are not the typical time period of the nighttime boundary layer and convective boundary layer, because the sunrise and sunset time in winter in Beijing is about 07: 00 and 19:00.Therefore, it is necessary to make a concrete analysis in view of different potential temperature profiles. The expression of "maximum potential temperature gradient method" in the first draft is indeed inappropriate, because of the influence of surface inversion and residual layer. As Seidel pointed out (Seidel et al., 2010), the vertical resolution of radiosonde data will affect the estimation of boundary layer height. So in the revised version, we collected and used more high-resolution radiosonde data.

As for the definition of inversion layer and residual layer, we also added in the introduction as follows:

Because of the Earth's rotation, the ABL presents strong diurnal variation, leading to the formation of many different layers in the boundary layer. The mixing layer accounts for a large proportion of the ABL in the deep convective boundary layer, and at present, the height of the mixing layer is equivalent to the height of the ABL. Pollutants emitted into the ABL can reach a certain height through turbulent vertical mixing processes (Emeis and Schäfer, 2006), making it possible to determine the ABL height from the concentration of pollutants. The top of the mixing layer exhibits capping inversion. Due to a change in the surface net radiation occurring at night, a stable boundary layer begins to form at night because of the cooling effect of the ground surface, and the surface inversion layer is nearest to the ground. The nocturnal stable boundary layer is often accompanied by a residual layer that maintains the characteristics of the daytime mixing layer (Stull, 1988)." (Page2 line7-15, in the revised paper)

The maximum potential temperature gradient of the first draft is indeed affected by the residual layer, which often appears bigger potential temperature gradient. At 08:00, there are often no obvious potential temperature gradients (For example 08:00 18 December, seen fig1). When the inversion layer caused by surface cooling is significant, there is the obvious change of potential temperature gradient, which can be regarded as the stable boundary layer height.

[Figure]

Fig. 1 Profiles of potential temperature and relative humidity at 08:00 18 December

In the revised paper Fig5.a b c (seen fig.2), PBL height can be determined by potential temperature gradient method. For Fig5b, besides the surface inversion on the ground, the height where potential temperature gradient reaches maximum is about 1000m, and there is also an obvious change of potential temperature gradient at 600m. At this time, combined with the change of relative humidity profile, the height of boundary layer is determined to be 680m by comprehensive analysis.

Therefore, in the revised version, the expression of the determining the boundary layer height by radiosonde is changed to:

"In this study, the level showing an obvious change in the potential temperature gradient and the profile of relative humidity were used to define the ABL height, as expressed by $H_\theta$ " (Page11, line13-14, in the revised paper)

[Figure]

Fig. 2 (Figure 5 in the revise paper)

*Refernce:*

*Seidel D J , Ao C O , Li K . Estimating climatological planetary boundary layer heights from radiosonde observations: Comparison of methods and uncertainty analysis[J]. Journal of Geophysical Research Atmospheres, 2010, 115(D16), doi: 10.1029/2009JD013680.*

**Major comments (b)**: In Sect. 3.4, the authors identify the PBLH as the maximum in the wind speed. While this method has been used before (as the authors note in the cited studies), it is only appropriate for identifying the PBLH when a low-level jet is present, which there is not during much of this study. Instead, the authors should use a more appropriate method for determining the PBLH from a WPR, such as Cohn and Angevine (2000) or Bianco et al (2002). It important to consider that WPR are often currently unable to measure the PBL height in the stable boundary layer due to the PBL being shallow (below the minimum range gate) and there is rarely a refractive index maximum signature. If a different more appropriate method is not used, then the WPR PBL height measurements should be removed from this study.

**Response:** According to the opinions of the reviewers, this maximum wind value can only be applied in the case of low-level jet structure, so we reanalyzed all the wind profiles of this process, and selected the profiles that have obvious "nose" structure or have reached the standard of low-level jet, so as to determine the PBL heights under this kind of wind profile. Here, we give the wind profiles of December 15, 17, 19 and 21 (all are local station time, seen fig.3). Although some cases did not meet the LLJ standard at some moments, it also had obvious "nose" structure. Below the height of the low level wind maximum is where affected by the ground friction, it can be concluded that the height of the low level wind extreme value at this time is the boundary layer height. The wind profiler radar can output the atmospheric refractive index structure parameter and determine the height of the boundary layer based on it. It is difficult to detect the stable boundary layer height because of the weak turbulence, so the boundary layer height based on refractive index structure parameter is mostly used in the convective boundary layer (Heo et al., 2002). However, atmospheric refractive index structure parameter was not used in this paper. Instead, wind profiles detected by WPR were used in this paper we, therefore, we still keep the results of wind profiler radar data, and gives the analysis results of the wind structure from low to high levels during the period of urban heavy pollution period.

[Figure]

Fig. 3 Wind profiles of December 15、17、19 and 21 (Figure 6 in the revised paper)

*Reference:*

*Heo B H , Jacoby-Koaly S , Kim K E , et al. Use of the Doppler Spectral Width to Improve the Estimation of the Convective Boundary Layer Height from UHF Wind Profiler Observations[J]. Journal of Atmospheric and Oceanic Technology, 2003, 20(3):408-424.*

**Major comments (c)** : This study could benefit from a discussion of the synoptic and mesoscale meteorology. Based on Fig. 2, it looks like there were cold fronts (strong NW winds) on both 12/15 and 12/22 which advected pollutants away and resulted in air quality. Between the fronts, the PM2.5 slowly increased as the air was stagnant (weak and variables winds in between) and pollutants emitted locally likely slowly built up. The wind direction also seems to be cyclical every day, perhaps this is in response to a local mountain-valley circulation around Beijing? This warrants more investigation. It appears that these factors, namely the stagnant air, dominate over the PBL processes in resulting in the poor air quality.

**Response:** We fully agree with the reviewer's analysis and added a paragraph and a reference to the revised paper as follows:

"From Fig. 2, we can see that there were cold fronts (strong NW winds) on both 15 December and 22 December, which advected pollutants away, resulting in good air quality. Between the fronts, PM2.5 levels slowly increased, as the air was stagnant (weak and variable winds in between), and the pollutants that were emitted locally slowly built up. The wind direction also seemed to be cyclical on each day in response to local mountain valley circulation around Beijing (Hu et al., 2005)." (Page6 line17-20, in the revised paper)

**Major comments (d):** While the authors provide numerous references early in the manuscript, there is little-to-no discussion of how the key findings and results compare to previous studies. For example, how does this haze episode compare to other poor air quality case studies in Beijing and other areas, considering both meteorology and boundary-layer processes? The authors should relate the hygroscopic growth to that observed in other studies (there are many). There have also been many other studies comparing lidar PBL heights with those from radiosonde and/or wind profilers, mainly focused on the convective boundary-layer where the top of the boundary-layer is clearly defined. These studies should be related to as well.

**Response:** The main purpose of this paper is to analyze characteristics of boundary layer structure during a red warning haze episode in Beijing, such a typical mega city, from the direct observation (radiosonde and tower) and remote sensing (lidar and wind profiler radar). We adopt the reviewer's opinions and analyzed the boundary layer height determined by lidar, radiosonde and wind profile radar in the polluted period. Therefore, the previous literatures on the boundary layer height have been added to the introduction. We also pointed out the shortcomings of previous studies and pointed out the relationship between this paper and some related studies in the revised paper:

"The ABL height is closely related to air pollution, but it is not the only factor that shapes air quality. Pollution conditions are also affected by wind speeds, emissions, chemical processing, etc. (Schäfer et al., 2006; Geiß, et al., 2017). Some works have compared ABL heights based on lidar and radiosonde data, and the correlations between them are stronger under unstable conditions (Emeis and Schäfer., 2006; Martucci, et al., 2007). However, for ABL heights determined from wind, relatively fewer studies have compared and analyzed the results of lidar and radiosonde tests applied during haze pollution episodes." (Page2 line15-20, in the revised paper)

"The Lidar results overestimate the ABL height at night (Quan et al., 2013). In this study the ABL height also increased due to the accumulation of pollutants during the period of heavy pollution." (Page18 line5-7, in the revised paper)

*Reference:*
*Schäfer K , Emeis S , Hoffmann H , et al. Influence of mixing layer height upon air pollution in urban and sub-urban areas[J]. Meteorologische Zeitschrift, 2006, 15(6):647-658.*
*Alexander Geiß, Wiegner M , Bonn B , et al. Mixing layer height as an indicator for urban air quality?[J]. Atmospheric Measurement Techniques, 2017, 10(8):2969-2988.*
*Emeis S , Schäfer K . Remote Sensing Methods to Investigate Boundary-layer Structures relevant to Air Pollution in Cities[J]. Boundary-Layer Meteorology, 2006, 121(2):377-385.*
*Martucci G , Matthey R , Mitev V , et al. Comparison between Backscatter Lidar and Radiosonde Measurements of the Diurnal and Nocturnal Stratification in the Lower Troposphere[J]. Journal of Atmospheric & Oceanic Technology, 2006, 24(7):1158-1164.*

**● Response to the Minor comments**

a) P. 1 line 13: 'Turbulent activities were great inhibited during haze pollution'. This must be rephrased, as the evidence in the paper does not support this statement. It is unclear if the haze suppresses turbulence, or if weak turbulence results in poorer air quality.

**Response:** Indeed, whether the turbulence affects pollutants or pollutants affects turbulence, the evidence in this paper is not very clear. This sentence has been changed to:

"Turbulence and pollutant concentrations are closely related during periods of haze pollution". (Page1 line14, in the revised paper)

b) P. 2, line 8: 'Please define all the acronyms, such as for URBAN, MIAGE, and SURF (similarly to how COST was defined and spelled out)'.

**Response:** The explanation of these abbreviations nouns is added in the revised paper. (Page.2 line22-27, in the revised paper)

c) Figure 1: Add a reference scale for distance on the plot. Currently, this map is not very informative in itself. It would be beneficial to add other details to the map relevant to the study (i.e., other important locations, elevation map, any significant pollutant sources, etc) if possible.

**Response:** We have redrawn this figure to make it more standard. First, the topographic map of Beijing and its surrounding areas have been added, and some important details have been added to the map, and the scale has been added. As shown in the Figure1, we marked the mountains, bays and other topography around Beijing, and marked the observation data during this period. In figure4, the east of Beijing is the Bohai Bay, so generally the easterly wind will bring water vapor to the Beijing area, which can help to explain some characteristics.

[Figure]

(a)                              (b)

Fig. 4 (Figure 1 in the revised paper) Local topography of Beijing and of its surrounding area (a). The locations of observation sites in Beijing (b): red circle: IAP (Lidar), blue circle: ZBAA radiosonde observation station, cyan circle: pollution observation station (OSCS) positioned approximately 2 km northeast of the Lidar. Beijing is a densely populated city covering an area of approximately 396 square kilometers.

d) P. 3 line 14: It is stated there are 15 platforms on the tower, but only 14 levels are subsequently listed. This inconsistency should be rectified.

**Response:** Thank you very much for your comments, and we are very sorry for this carelessness. We have modified this mistake. (Page4, line3, in the revised paper)

e) P. 4 line 16: List out the six air pollutants measured.

**Response:** We have listed out six types of pollutants here and modified it in the paper. (Page5, line1, in the revised paper)

f) Table 1: What is the difference between 'heavy polluted' and 'serious polluted'? Describe in the text these categories (and all the other ones). It is unclear which category is worse. These are later defined in Table 2 at the end of the manuscript, but these definitions should be moved to the discussion of table 1.

**Response:** We have adopted the comment by reviewer, and have moved the definition to the discussion of Table 1. (Page5 table1, in the revised paper)

g) P. 5, line 3: Here, the authors claim that the lack of diurnal variation of temperature and RH is due to heavy pollution. This is plausible, but another reason is more likely. The high RH and low visibility, secondarily combined with the low wind speed, all indicate that fog or low stratus is present. The fog or stratus itself would greatly reduce insolation and daytime heating. This would effect would likely dominate over aerosol effects in reducing solar radiation reaching the surface. The discussion here needs to be modified accordingly. It would also be useful to include a plot of cloud cover somewhere.

**Response:** Thank you for your help with the modification explanation of the temperature and RH diurnal variation characteristics due to the heavy pollution. We have added the MODIS cloud figures in this paper (Figure.3 in the revised paper), and further analysis of satellite cloud images during this period shows that:

"pollution process was indeed accompanied by fog, while pollution formed in the south-central area of Hebei Province on 15 December 2016 and then spread across the whole Beijing-Tianjin-Hebei area on 18 December. Stratiform clouds appeared in areas surrounding Beijing on 21 December, but due to the high concentrations of pollutants (PM2.5 values approaching $400\,\mu g$ m-3, mixed fog and haze appeared in Beijing. During the day, pollutants can

scatter more solar radiation while the ground receives less solar radiation, leading to the suppression of diurnal variations in temperature and relative humidity on the ground. (Page6, line6-11, in the revised paper)

[Figure]

(a) (b) (c) (d)

Fig. 5. (Figure. 3 in the revised paper) MODIS image of Beijing-Tian-Hebei area from 14-23 December 2016. (a)15 December 2016, (b)18 December 2016, (c)21 December 2016, (d)22 December 2016

h) P. 5 lines 6-7: RH cannot be used to quantify the total amount of water vapor in the air, as RH is highly-dependent on the temperature. The increase in moisture discussed here may simply be due to a lower temperature, not an increase in water vapor content. Please use mixing ratio or another conserved quantity.

**Response:** Because relative humidity is dependent on temperature, so that this paper compared relative humidity on the condition of the same temperature. Ground observation data shows that the temperature at 00:00 17 December and 00:00 20 December were all -4℃, but the relative humidity at 00:00 20 December was 100%, far more than it at 00:00 17 December(74%).Since the temperature was the same, we infer that there was an increase in water vapor at 00:00 20 December.

| Time | Temperature (℃) | RH (%) | Wind speed (m s-1) | Wind direction (degree) | Visibility (km) |
|---|---|---|---|---|---|
| 2016/12/17 00:00 | -4 | 74 | 2 | 350 | 4 |

| 2016/12/20 00:00 | -4 | 100 | 1 | 11 | 0.2 |

i) Sect 3.2: Rename this section "Lidar observed boundary layer heights", as the WPR and radiosonde and not discussed in it. These paragraph needs to be rewritten and better organized. Start by explaining why the extinction profile can be used to identify the ABL height, then lead into three different methods that can be used that will yield different estimates, but also each have their own limitations. Describe each method separately, as currently the description of all three is spread throughout the entire section.

**Response:** Thank you for your comment. We have modified it to "Boundary layer heights observed by Lidar". In addition, we have rewritten this paragraph as follows:

"The most basic definition of the ABL height is the height at which the influence of the Earth's surface on the lower troposphere disappears. This influence applies not only to conventional meteorological elements but also to turbulence quantities and even more for substances in the atmosphere such as aerosols, water vapor and nonreactive tracer gases (Seibert et al., 2000). Levels of various pollutants and water vapor in the ABL are much higher than those found in the free atmosphere, and therefore, there is often an obvious aerosol concentration gradient between the boundary layer and the free atmosphere. The extinction coefficient reflects the degree of aerosol particle scattering from lasers in the atmosphere (Boers and Eloranta, 1986). Thus, the ABL height can also be estimated from the extinction coefficient gradient. We used three popular methods---the gradient method (Lidar_gra) (Flamant et al., 1997), the standard deviation method (Lidar_std) (Hooper and Eloranta, 1986) and the wavelet method (Lidar_wav) (Cohn and Angevine, 2000; Davis et al., 2000; Brooks, 2003) ---to extract boundary layer heights from extinction coefficients. The ABL height determined by Lidar is represented by Hc. In this study, the Lidar_gra method applies the height of the atmosphere at which the gradient of the Lidar extinction coefficient reaches its most negative value. The standard deviation of the extinction coefficient reflects the degree of Lidar echo signal dispersion at different heights. The top of the planetary boundary layer constitutes the intersection between air in the boundary layer and the free atmosphere, which leads to a strong signal change at the top of the boundary layer. We define the height of the maximum standard deviation of signals as the ABL height. The Lidar_wav method can also be used to detect abrupt changes in signals, so we use the Haar wavelet and take the height at which the wavelet coefficient is at its highest value as the height of the ABL. These methods are used
to find the abrupt change in the extinction coefficient occurring at the top of
boundary layer, though they present their own limitations.
Generally, the atmospheric boundary layer can be divided into a daytime
convective mixing layer and a nighttime stable boundary layer. In the morning, the well-mixed convective boundary layer (CBL) is growing and often reaches its maximum height in the early afternoon. In the afternoon, the CBL gradually transforms into a neutral boundary layer. Figure 4 illustrates the evolution of ABL heights measured with Lidar, WPR and radiosonde tools."
(Page8-9, in the revised paper)

j) P. 7, line 10: What is the dilation of the Haar wavelet? The determined ABL height

has been shown to be sensitive to this in many previous studies.

**Response:** The selection of dilation of the Haar wavelet is the key in this wavelet transformation method, and Brooks (2003) proposed that the best selection of dilation parameter *a* should be equal the transition zone range of the ideal profile. We take *a=6* as the dilation of continuous Haar wavelet transformation in this paper, and PBL heights calculated based on it coincide with the edge regions where distinct changes of extinction coefficients exist. We have added the following in the revised paper:

"(continuous Haar wavelet transformation was used in this paper, taking dilation paramet a=6)." (Page 8, line 11 in the revised paper)

*Reference:*
*Brooks I M . Finding Boundary Layer Top: Application of a Wavelet Covariance Transform to Lidar Backscatter Profiles[J]. Journal of Atmospheric and Oceanic Technology, 2003, 20(8):1092--1105.*

k) Figures 3 and 4: Why is extension near the surface so small? I would expect the extinction profile to be nearly uniform (most of the time, except in the presence of clouds) throughout the entire PBL. This leads me to believe that the overlap correction for the lidar is not correctly determined and/or applied. Also, in calculating the extinction coefficient, is attenuation considered? Also, please state if time here (and throughout the manuscript) is local time or GMT.

**Response:** Mainly because the blind zone of the radar has a thickness of several tens of meters, and the measurement in the near-surface layer is not reliable. We have stated the time used in this paper is local station time in Section2: Observation sites, instruments and data "We use local station time in this work, and the observational instruments and data employed are as follows:……" (Page3 line28, in the revised paper)

l) P. 8 line 8: What is meant by 'transformation zone'? This is not a commonly used term. m) P. 8 line 10: This reasoning is insufficient to discount the lidar observations, especially given following concerns with the PBLH determined from both the radiosonde and wind profiler. If the extinction is plotted on a logarithmic scale (which is more appropriate given the large dynamic range that it can vary), a real gradient associated with the PBLH determined by the lidar would likely be more apparent. Hints of this gradient are apparent on the plot now during these clean periods at around 250m on 12/15 and 12/22.

**Response:** thanks to the reviewer, "transformation zone" should be "transition zone", and have rewritten this expression into:

"the aerosol concentration was low and the extinction coefficient derived from the Lidar system

displayed no obvious signs of decline from the ground to the upper height" (Page10, line8-9, in the revised paper).

In this paper, the extinction coefficient is lower and the height where holding big extinction coefficient gradient is relatively low of clean days.

n) P. 8, line 23: The fact that the PBL was statically stable is unsurprising given the timing of the radiosonde profiles at 08:00 and 20:00 local time, which are respectively about 30 min after sunrise and 3 hours after sunset during the experimental period. Thus, a nocturnal inversion would barely erode by 08:00 (if at all, depending on the energy balance as insolation is small) and a nocturnal inversion would have formed by 20:00. Due to the timing, these profiles are not representative of conditions during the day when pollutants are actively mixed. I suspect that if the profiles were during the midday (noon local time or early afternoon) the profiles would indicate instability and mixing. This must be discussed at the very least.
**Response:** Thank you for the comments pointed out by the reviewers, we adopted this discussion in the text. However, due to the limitation of radiosonde in Beijing, only profiles at 08:00 and 20:00 are available. By adding the observation profiles of the daytime of the tower, the typical convective or nocturnal boundary layer characteristics are only analyzed by means of the tower observation.

o) P. 8, line 28-29: Based on this description (and the profiles), it seems as though the top of the stable boundary layer where mixing is occurring is at 100 m (not 700 m), while the top of the decoupled residual layer is at 700 m.
**Response:** Thanks to the reviewer for pointing out our mistakes, At this time, the potential temperature between 100-600m is not almost unchanged, or it shows an obvious inversion layer, so we revised the relevant expression in the paper:

"Due to the cooling effects of surface longwave radiation, the ground inversion layer frmed from the surface at a depth of approximately 100 m. At this time, the potential temperature gradient underwent an obvious change at 600 m. The inversion intensity levels below 600 m was weaker, and the height Htheta was approximately 690m." (Page11, line30-32, in the revised paper)

p) P.9, line 5: Why would an easterly wind advect water vapor? This is unclear and not supported by the data. A map of the water vapor field at the time would be needed to support this.
**Response:** Thank the reviewers for their comments. It is indeed our negligence. By giving topographic map (Figure1 in the revised paper), it can be seen that the east part of Beijing is the Bohai Bay. Generally, the easterly wind, especially the southeast wind, can bring abundant water vapor to Beijing.

q) P. 9 line 6: The authors should provide a reference here for hygroscopic growth, as it has been detailed in numerous prior studies.

**Response:** P.9 line6: According to the reviewer's suggestion, here we have added 3 references: Svenningsson et al. (1992), Chuan (2003) and Pan et al. (2009).

r) P. 9 line 8: The profile still appears stable in panel c) as the inversion is not completely eroded yet with potential temperature increasing from the surface upwards, thus the ground inversion is still apparent (although weaker) than in panels a) and b).

**Response:** We adopted this discussion and we have deleted the expression "The ground inversion layer has disappeared".

s) P. 9 line 11: Why is H_theta at 604m? By eye, the maximum in the gradient of the potential temperature (and humidity) is near 350 m.

**Response:** Initially, because of the low vertical resolution of our radiosonde data. Based on the reviewer's opinion, we used the high vertical resolution radiosonde data and the new calculated $H_\theta$ was about 410m. (Page12, line12, in the revised paper)

t) Figure 4: The tick marks on the right-most y-axis in panel c) are not aligned with all the other tick marks, which is confusing and deceptive. Also, why do all the profiles start at around 50 m? Is this height above ground level or height above sea level?

**Response:** We redrew Figure4 (Figure.5 in the revised paper) and made sure that the tick marks on the right-most y-axis in panel c) are aligned with the other tick marks. The height is above the ground level, and first sounding data is basically from 30m.

u) P. 9 line 13: Rephrase the sentence "At the inversion layer, it was easier to appear the larger value of the potential temperature gradient", as its meaning is unclear.

**Response:** we have rewritten this sentence and made it more clear:

The potential temperature gradient at the inversion layer is generally larger. (Page12, line16, in the revised paper)

v) P. 9 line 16: Again, please provide evidence to support this statement. Perhaps in Fig. 4 a few panels could be added (and a supporting discussion in the text) that show profiles where there is no inversion layer apparent (or it is weak).

**Response:** According to the reviewer's opinion, considering that this paper is only a case study of one air pollution process, there is not enough evidence to support the sentence "However, when the atmosphere was clean, the aerosol concentration was obviously reduced and the inversion layer was not so significant.", so we remove this sentence in the text.

w) P. 11 line 4: The equation for u_* is incorrect. The u'w' and v'w' quantities should both be squared. Make sure it is calculated correctly in Fig. 6, b as well.

**Response:** The calculation of friction velocity in Fig. 6 is corrected. Thank you for your reminder. We revised the expression of friction velocity correctly. (Page15 line5, in the revised paper)

x)    Fig. 7: Please make the colorbar for the wind direction (in leftmost panel) circular. With the current colorscale, wind direction at 359 deg is red and 1 deg is blue, even though there's little difference in the wind direction. It would also be beneficial to provide plots from daytime conditions when the PBL is well-mixed during the pollution episode, as both of the selected time periods are near midnight

**Response:** Based on your comments, we have redrawn these figures to make the wind direction clear, and have deleted the description of $\overline{w'^3}$ and added the plots from daytime comparison between clear and pollution day. (Figure9 and Figure10 in the revised paper):

"In the daytime on 21 December 2016 (see Fig. 10), the $PM_{2.5}$ concentrations reached roughly $400\,\mu g$ m$^{-3}$, wind speeds were low, TKE remained zero value, and the potential temperature observed from the tower during the period of pollution basically denoted neutral stratification. The sensible heat flux was positive, but the value was basically measured as 0.02 K m s$^{-1}$. At noon on 22 December, when the weather had improved, wind speeds were clearly higher. TKE still reached a maximum at 47 m. The influence of the urban canopy was stronger at heights of below 47 m. Unlike on the polluted day, levels of sensible heat flux were higher at this time, and the lower layer reached a value of 0.1 K m s$^{-1}$. The tower observation data clearly show that levels of sensible heat flux decreased significantly in the daytime during the haze episode because of the higher levels of solar radiation scattering by particles." (Page17 line1-8, in the revised paper)

y) Table 2 (and its discussion): There appears to be little difference between slightly, moderately, heavily, and seriously polluted conditions. There is no clear trend in the listed variables with increasing pollution. I suspect that the calculated small differences are not statistically different between categories, except compared with 'good' air quality.This is unsurprising, given the meteorology remains roughly constant during all periods between times when the air quality is 'good' when pollutants slowly build up. This should be discussed

**Response:** Thanks to the reviewer for his valuable comments, wo use the ABL height "heightened slightly", and added a sentence in the text:

  "Since the data examined in this study apply to only one period of heavy pollution, the minor differences observed may not be statistically different across categories except when compared to "good" air quality conditions." (Page18, line 10-11, in the revised paper)

z) P. 15 line 6: Again, there is no evidence in the manuscript that north-easterly winds brought pollutants and water vapor to Beijing. Based on Fig. 2, it looks like this first increase in pollution was the first night when winds were light, thus pollutants likely

emitted locally were not dispersed and built up quickly in a shallow layer near the surface.

**Response:** P.15 line6: Thank the reviewers for their comments. At first, we analyzed the 10m wind field data from FNL reanalysis data. The convergence zone of northeast and southeast winds in the east part of Beijing is shown in the yellow dotted line frame. Later, the 10m wind field data map was not added due to the use of ground observation data and wind profile radar. So, we revised the relevant expression in the paper.

[Figure]

Fig.6.10m wind field from FNL reanalysis data (08:00 17 December)

aa) P. 15 line 13: Rephrase the sentence 'The turbulent fluxes…radiation from the ground", as its meaning is unclear

**Response:** Based on your comment, we have rewritten this sentence to make it clear "Turbulent fluxes varied very little with altitude, but the sensible heat flux measured at night was slightly positive close to the Earth's surface, indicating that the cooling effect is inhibited by long-wave radiation from the ground." (Page20 line15-17, in the revised paper)

bb) P. 15 line 15: How is turbulence suppressed when the sensible heat flux is positive? Normally a negative heat flux with cooling near the surface will strengthen the stability limiting turbulence, while a positive heat flux weakens stability

**Response:** Yes, our high-resolution 7-layer turbulence instrument observation shows that the sensible heat flux on the ground was positive at night during the pollution period. First of all, at night, due to the presence of pollutants and more vapor, the long-wave radiation cooling effect from the ground was inhibited. Furthermore, in a city with large population, such as Beijing, the building heat storage and anthropogenic heat of the city also made it possible that the surface sensible heat flux

appeared positive at night.

**2. Response to RC2**

**Response to the RC2**

**Comments:**

The authors have addressed my comments satisfactorily. I think the manuscript is acceptable for the publication.

**Response:**

Thank you very much for your positive comments on our paper and think it is acceptable for publication. The revised paper was sent to you as an attachment.

**3. Response to RC3**

Dear reviewer,

Thank you very much for your comments on our paper! Your review and comments are very helpful to our paper, and we have learned a lot from them. We reply to your comments one by one as follows.

**Comments 1:** Only in the beginning, a definition of the ABL and the different layers should be helpful, e.g. Mixing Layer, Surface Inversion Layer, Capping Inversion, Convective Mixed Layer, Residual Layer.

**Response**:We have added the definition of the Mixing layer, surface inversion layer, capping inversion, convective mixing layer and residual layer in the latest revised paper:

"Because of the Earth's rotation, the ABL presents strong diurnal variation, leading to the formation of many different layers in the boundary layer. The mixing layer accounts for a large proportion of the ABL in the deep convective boundary layer, and at present, the height of the mixing layer is equivalent to the height of the ABL. Pollutants emitted into the ABL can reach a certain height through turbulent vertical mixing processes (Emeis and Schäfer, 2006), making it possible to determine the ABL height from the concentration of pollutants. The top of the mixing layer exhibits capping inversion. Due to a change in the surface net radiation occurring at night, a stable boundary layer begins to form at night because of the cooling effect of the ground surface, and the surface inversion layer is nearest to the ground. The nocturnal stable boundary layer is often accompanied by a residual layer that maintains the characteristics of the daytime mixing layer (Stull, 1988)." (Page2 Line7-15, revised paper).

**Comments2:** Usually the ABL is the turbulent layer with winds influenced by the earth surface. Within this definition the Wind Profile Radar (WPR – the full word should not only be used in the abstract) gives the best result of the ABL height. In the air quality community the mixing layer and inversion layers are more common to use in the context of air pollution concentrations. These terms are explained at the beginning of chapter 3.2. But this could be done some more concisely., perhaps extracting one day of Fig. 3 and explaining it. In Fig. 3 the heights Hc and Hu should be marked which are reffered to later.

**Response**:We have added the full word of WPR (wind profile radar) not only in the abstract, and wind profile radar (WPR) appeared first in the introduction (Page2 line19, revised paper) and the all the abbreviations are used later in the paper. We recalculated the ABL height determined by WPR for distinct "nose" profiles. And the new ABL heights are exhibited in Fig.4 in latest revised paper. In addition, the heights Hu and Hc are marked as shown in revised paper. The definition of mixing layer and inversion layers have also been added in the revised paper. (Page2 Line7-10, revised paper)

[Figure]

Fig.4 Temporal and spatial variations in the extinction coefficient (shaded, unit: $km^{-1}$) from 14 to 23 December 2016 and ABL heights (m) determined with different instruments. The red line (Lidar_gra), grey line (Lidar_std), and purple line (Lidar_wav) represent ABL heights determined by the Lidar using the gradient method, the standard deviation method, and the wavelet method, respectively. White points: ABL height determined by radiosonde; Five-pointed star: ABL height determined by WPR. It should be noted that the blank part of the extinction coefficient can be attributed to a technical failure, and lidar data for 11:00 on December 19 to 09:00 on December 20 are missing.

**Comments3:** Table 2: The first column shows the air quality, I think these are PM2.5 concentration ranges. It should be mentioned. There is a general influence of humidity. But not so clear. That is why the humidity depends on the origin of the advected air (more wet or more dry). Important is the decrease jump at the inversion layer, shown in Fig. 4 and not the abolute value of the humidity. Page 13 line 5… "Hc is even heightened lightly but Hu (not Hc) reduces by…"

**Response:** For table2, we have explained the $PM_{2.5}$ concentration of the corresponding air quality in the table1 head, and we also accept the idea of reviewer to reexplain it in the table2 header (Page19, Table2, revised paper). As for page13 line5, "Hc is even heightened slightly but Hu (not Hc) reduces by…" thank you very much for your carefulness and we have corrected this error (Page20, Line12, in the revised paper).

**Comments4:** In the conclusion the role of the mixing layer and the inversion layers should be higlighted in the context with air pollution concentrations. And these layers are best determined by radiosonde soundings and lidar. The tower measurements are also helpful to determine the surface inversions height, if the inversion layer is lower than the tower height. With the Wind Profile Radar (WPR) the height of the Atmospheric Boundary Layer (turbulent) can be well determined. But for the air pollutant concentrations (PM 2.5) the inversion Layers and the Mixing Layer are relevant which is shown in this paper.

**Response:** In revised paper, we have highlighted the relationship between mixing layer, inversion layer and pollutant concentration in the conclusion.

"The inversion layer is closely related to the concentration of pollutants. Pollutants emitted in the ABL generally accumulate under the inversion layer. The inversion layer's height decreased significantly during the pollution period, and the lowest value was measured at below 500m." (Page20, Line8-10, revised paper).

**4. Response to the Editorial corrections**

a) The manuscript needs heavy editorial corrections thoughout.
**Response:** We have carefully revised and edited the full text of the paper and many details have been rewritten.

b) P. 3, line 12: 49 m above sea level is listed twice in this sentence. Remove one instance.
**Response:** One of the two "49m" was removed. (Page4 line1, in the revised paper)

b) P. 3, line 13: Why is 'August 1979' here?
**Response:** P.3 line 13: Our original intention is that this tower was established in 1979 and has been revised in paper.

[revised manuscript text omitted]